# Potential of artificial intelligence in reducing energy and carbon emissions of commercial buildings at scale

Chao Ding ●[1], Jing Ke ●[1], Mark Levine[1], Jessica Granderson ●[1] & Nan Zhou ●[1] ✉

Artificial intelligence has emerged as a technology to enhance productivity and improve life quality. However, its role in building energy efficiency and carbon emission reduction has not been systematically studied. This study evaluated artificial intelligence's potential in the building sector, focusing on medium office buildings in the United States. A methodology was developed to assess and quantify potential emissions reductions. Key areas identified were equipment, occupancy influence, control and operation, and design and construction. Six scenarios were used to estimate energy and emissions savings across representative climate zones. Here we show that artificial intelligence could reduce cost premiums, enhancing high energy efficiency and net zero building penetration. Adopting artificial intelligence could reduce energy consumption and carbon emissions by approximately 8% to 19% in 2050. Combining with energy policy and low-carbon power generation could approximately reduce energy consumption by 40% and carbon emissions by 90% compared to business-as-usual scenarios in 2050.

Climate change is a critical theme and challenge in the past and current centuries. Many regions of the world have experienced extreme weather events such as floods and heat waves, and these events have been happening more frequently in recent years. To limit global warming, the Paris Climate Agreement sets a 1.5 °C global warming target by reducing energy consumption and carbon emissions[1]. Governments worldwide have already established ambitious climate targets to address this goal. For example, the United States is targeting a 50–52% greenhouse gas (GHG) pollution reduction from 2005 levels in 2030[2], China announced[3] an action plan for $CO_2$ peaking before 2030, and the European Union proposes to cut GHG emissions to at least 55% below 1990 levels by 2030[4].

Meanwhile, the global urban population has experienced rapid growth since 1950. In 2018, ~55% of the world's population lived in urban areas. This number is expected to increase to 68% by 2050, according to a United Nations prediction[5]. Due to this rapid urbanization, there will be considerable new construction. The world's building stock is expected to double by 2060, which is equivalent to building an entire New York City every month for the next 40 years[6].

According to the US Energy Information Administration (EIA), the building sector accounted for 39% of primary energy consumption in the United States in 2011[7]. Therefore, to support energy efficiency and carbon reduction targets, it is crucial to study buildings.

With the rapid development of computer technology, artificial intelligence (AI) is becoming increasingly accessible for different domains and applications. It "solves complex cognitive problems commonly associated with human intelligence, such as learning, problem-solving, and pattern recognition"[8]. As of February 2020, 50 countries had announced national AI strategies[9].

AI has been widely adopted in different application domains, such as computer vision, robotics, natural language processing, and machinery. Recently, it has also been used to improve energy efficiency and reduce carbon emissions in the building[10–12] transportation[13–15], and industry sectors[16–18]. Some popular research directions include smart control, system diagnostics, occupancy behavior, load prediction, and demand response.

However, the energy-saving potential of AI in buildings has not been thoroughly understood. Existing literature primarily focuses on specific aspects of building performance such as building design[19,20],

[1]Energy Technologies Area, Lawrence Berkeley National Laboratory, One Cyclotron Road, Berkeley, CA 94720, USA. ✉e-mail: nzhou@lbl.gov

construction[21–24], operation and control[25–28]; the research indicates a diverse range of potential savings, spanning from 2 to 60%. Moreover, a systematic and standard method for AI savings quantification is still lacking. The mechanism of how AI could best be used to reduce energy consumption and carbon emissions in buildings is also not clear.

Buildings are complex systems, with thousands of components, such as walls, windows, and HVAC and lighting systems. Building constructions usually involve planning, analyzing, developing and constructing, each requiring substantial knowledge, investment, and labor. Building constructions often pose potential threats to the health and safety of construction workers. AI has the potential to reduce costs across various stages of the construction process, mitigate risks, and enhance health and welfare benefits[22–24,29–33]. Moreover, the interactions between building occupants and building components are non-linear and difficult to capture using traditional rule-based control algorithms. With advanced AI algorithms such as deep learning and reinforcement learning, the AI model can itself learn from operational data and evolve itself with continuous live data to optimize objective functions and improve performance.

What is the future of high-efficiency buildings in the United States? How can AI influence building design? If we assume that AI reduces the cost of implementing energy efficiency by 5%, how much additional energy and carbon benefit could be achieved? Instead of concentrating on a specific AI technology, the objective of this study is to explore the potential impact of AI on enhancing energy efficiency and reducing carbon emissions in commercial buildings at scale. Furthermore, the study aims to propose a systematic approach that can be applied to quantify the benefits of AI in various building types beyond commercial buildings. The paper is organized as follows. Result section proposes a four-key structure to systematical decouple and evaluate the theoretical maximum energy-saving potentials across an individual building's lifetime; and quantifies the impact of AI at scale by developing energy efficiency technology adoption and building stock modeling. Discussion section discusses future research directions and summarizes the conclusions. Finally, Methods section describes the methodologies in details.

## Results

### AI's impact on energy and emission reductions

To understand the areas in which AI can contribute to reducing energy consumption, it is first necessary to understand the average building energy efficiency level and the current best practices.

According to the EIA's Commercial Buildings Energy Consumption Survey (CBECS) in 2012[34], office buildings are the most common type and account for the highest electricity consumption (20%) among all commercial buildings. The median US office building energy use intensity (EUI) is 167 kilowatt-hours per square meter (kWh/m²) (53 thousand British thermal units per square foot, kBtu/sf), which is

considered to be the baseline office building energy consumption level ($EUI_{base}$) in this study. Based on a review of 67 low-energy verified buildings in the United States, the median of the gross EUI is 57 kWh/m² (18 kBtu/sf)[35]. The difference between the average and the best practice (high-energy efficiency building, HEEB) is assumed to be the technical building energy saving, which may be attributed to several factors.

We break down the building energy saving into four key categories: (1) equipment, (2) occupancy influence, (3) control and operation, and (4) design and construction.

The energy consumption in office buildings is dominated by medium office buildings, accounting for 70% of the total[34]. To demonstrate our methodology, a medium office building was selected as an example for the following analysis. Annual building energy consumption of a prototype medium office building was modeled using the US Department of Energy's (DOE's) EnergyPlus simulation tool. The prototype medium office model is defined by ASHRAE standard 90.1[36], which has detailed building characteristics such as geometry, thermal properties, HVAC system, occupancy, and more. The geometry information, as well as other key assumptions of the baseline EnergyPlus model setting, is summarized in Supplementary Table 1. The DOE's prototype building models have undergone rigorous calibration using real data, making them highly reliable. These calibrated models have been used in the evaluation of building energy codes and the formulation of code amendments by the DOE's Building Energy Codes Program[37].

Four representative cities were simulated in detail, to consider building energy consumption in different climate zones. Table 1 shows the baseline medium office energy consumption of each selected city. The electricity use breakdown can be found in Fig. 1. It was assumed that natural gas is used to provide hot water and heating.

EnergyPlus building energy models were developed to consider different design variations among the proposed four key categories.

The Pacific Northwest National Laboratory (PNNL) conducted a comprehensive analysis to study the impact of control measures on US commercial building energy savings[27]. Thirty-four energy efficiency measures in nine prototypical buildings were modeled using Energy-Plus. Sixteen US climate regions were considered and weighted according to the EIA's 2012 CBECS data. The study assumed three different penetration scenarios (inefficient, typical, and efficient) and calculated the national total energy savings of the different building types. The research showed that the national typical total energy saving from controls is 27.2% for a medium office.

We developed eight cases to evaluate the energy-saving potentials from equipment efficiency improvement. Supplementary Table 2 summarizes the energy savings from each case among the four different climate zones. Based on the energy performance of the baseline equipment and commercially available best practices, conservative equipment efficiency improvement assumptions are made in the analysis to consider the uncertainty and technology applicability across different climate zones. Cases 1–3 show the energy-saving potentials from the HVAC system. The baseline system provides cooling using a two-speed direct expansion cooling coil and heating using a natural gas furnace inside the packaged air conditioning unit. Case 1 increases the cooling efficiency by 20%. Case 2 increases the heating efficiency by 12%. Case 3 integrates both cooling and heating efficiency improvements. Cases 4 and 5 show the energy-saving potentials from the lighting system. The baseline average lighting power density (LPD) is 10.76 watts per square meter (W/m²). Cases 4 and 5 reduce the LPD by 15% and 21%, respectively. Cases 6 and 7 show the energy-saving potentials from the electric equipment and plug loads. The baseline average equipment power density (EPD) is 8.07 W/m² with two elevators (32,433 W). Cases 6 and 7 reduce the EPD by 10% and 20%, respectively. Case 8 combines the energy-saving

**Table 1 | Simulated total energy consumption of the baseline medium office buildings**

| Baseline | Climate zone | Electricity (kWh/m²) | Natural gas (kWh/m²) | Total (kWh/m²) |
|---|---|---|---|---|
| Honolulu | 1A | 158 | 4.0 | 162 |
| Los Angeles | 3B | 127 | 4.6 | 132 |
| Baltimore | 4A | 140 | 16.2 | 155 |
| Buffalo | 5A | 141 | 26.8 | 168 |

Annual energy consumptions of baseline medium office buildings from four representative US climate zones defined by the International Energy Conservation Code (IECC) standard were simulated in EnergyPlus based on key input assumptions from building energy efficiency standards[36]. The details are summarized in Supplementary Table 1. The results were verified using CBECS 2012 data[34]. Figure 1 shows the electricity breakdowns of each representative city. Note that the total electricity consumption, as well as the energy use breakdown percentage, vary by climate. It is necessary to conduct whole building energy simulation for each climate zone.

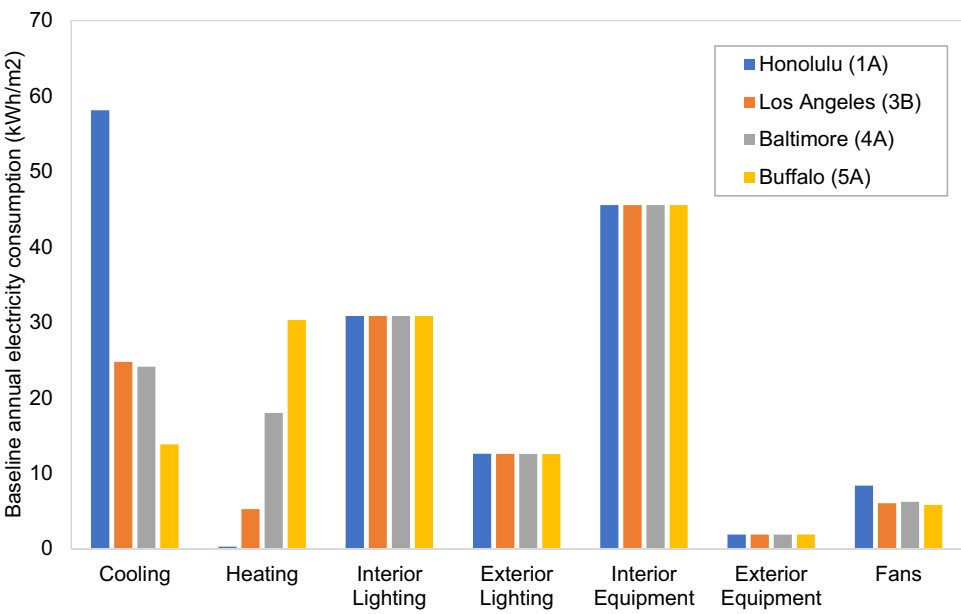

**Fig. 1 | Baseline annual electricity consumption, by use and climate zone.** The four selected cities represent four typical climate zones in the United States. Sources: authors' calculation based on annual building electricity consumptions of the baseline EnergyPlus model described in Supplementary Table 1.

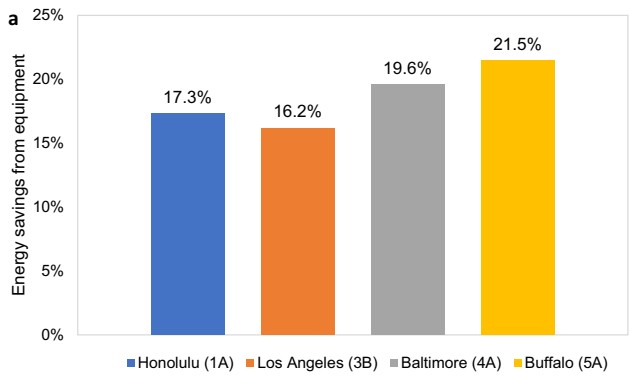

**Fig. 2 | Integrated energy-saving potentials based on annual building energy simulations. a** Savings from equipment efficiency improvement. **b** Savings from building design and construction. The four selected cities represent four typical climate zones (CZs) in the United States. Sources: authors' calculation based on annual building energy simulations of different cases described in Supplementary Tables 2 and 3.

measures of Cases 1–7. Case 9 considers the replacement of packaged air conditioning units with heat pumps for space heating in addition to Case 8. Supplementary Table 2 shows the integrated energy-saving potentials. The total energy savings from equipment efficiency improvement are 11.5-17.3%.

Regarding building design and construction, nine cases were considered to evaluate the energy-saving potential. Supplementary Table 3 summarizes the energy savings from each scenario among the four different climate zones. Cases 1–3 show the energy-saving potentials from building orientations. The baseline orientation is north. The other three orientations were achieved by rotating the building by 90° (east), 180° (south), and 270° (west). Based on this analysis, we found that orientation has a limited impact on annual building energy consumption in three different climate zones. The best orientation is north. Cases 4 and 5 show the energy-saving potentials from building envelopes. Case 4 implements high insulations for external walls, slabs, roofs, and windows. Supplementary Table 4 shows in Case 4 U-factors of different envelope components compared with the baseline model. Using high levels of insulation can save 3.2-6.7% of total building energy consumption. Case 5 increases the infiltration by ~60% to match the ASHRAE 90.1-2016 prototype

medium office building level. Cases 6–8 studied the energy-saving impact from window-to-wall (WWR) ratio. The baseline average WWR is 0.33. Case 9 combines all three energy-saving measures (orientation, insulation, and WWR). Supplementary Table 3 shows the integrated energy-saving potentials. The total energy savings from building design and construction are 5.9-9.1%.

Energy-related occupant behavior in buildings includes open/close windows, switch/dim lights, turn on/off lights and plug loads, turn on/off HVAC and adjust thermostat, and more. Occupants can interact with building energy systems (HVAC, windows, lights, and plug-in equipment). Based on the literature review, the integrated energy-saving potential between wasteful and austerity is 15-20%[38–40].

Integrating the four climate zones shown in Figs. 2 and 3 (based on the work described herein) summarizes the energy-saving potentials of a typical medium office building in the United States. The energy-saving potentials across the building's lifecycle vary across the four different climate zones for each category. Noted that the saving potentials are maximum technical potentials based on energy simulation results, and that it may require a large amount of time and effort from architects and engineers to optimize the entire building system

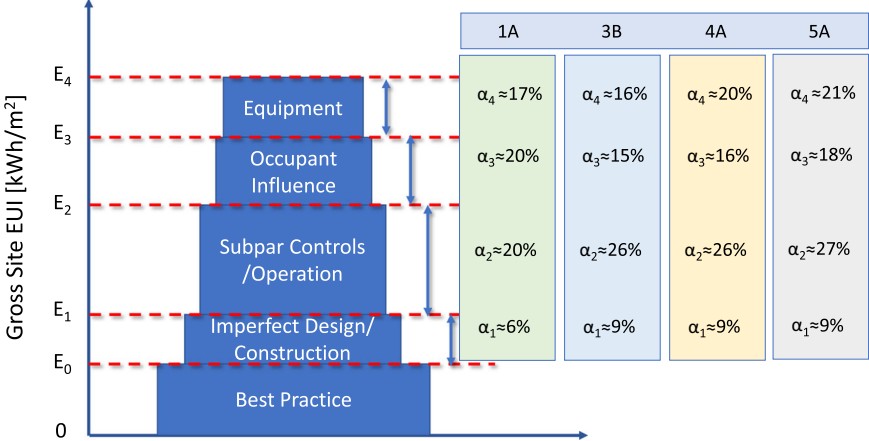

**Fig. 3 | Integrated technical building energy-saving potential of a typical medium office building in the United States.** We broke down the building energy saving into four key categories $\alpha_1$–$\alpha_4$ are the saving potentials (%). $E_0$–$E_4$ represent different gross site energy use intensity (EUI). $E_0$ represents the median of US office buildings[34]; $E_4$ is the median of zero energy verified building in the US[35]. The selected climate zones (CZs) represent the top four regions (Honolulu: CZ 1A, Baltimore: CZ 4A, Buffalo: CZ 5A, and Los Angeles: CZ 3B) defined by the International Energy Conservation Code (IECC) based on a number of office buildings from CBECS 2012 data[34,59].

**Table 2 | Examples of AI applications in different categories**

| | Problem | Example AI application | Energy-saving potential | Cost-saving potential |
|---|---|---|---|---|
| Equipment | Aging equipment | Failure detection, predictive maintenance[53] | √ | |
| Imperfect design | MEP clash | BIM model checker[22] | | √ |
| | Complex design | Architectural design optimization[32] | √ | √ |
| Imperfect construction | Schedule management | Project schedule optimizer[33] | | √ |
| | Envelope leakage, infiltration | Drone/robotic construction[54] | √ | √ |
| | | 3D concrete printing/digital construction[55] | √ | √ |
| | Workforce/safety management | Safety sensors, image recognition[56] | | √ |
| Subpar controls/operation | System malfunction/failure | Fault detection diagnostics (FDD)[57] | √ | √ |
| | Optimal control | Model predictive control; deep learning; reinforcement learning[11] | √ | |
| Occupant influence | Thermal comfort | Smart occupancy sensor, behavior prediction[38–40] | √ | |
| Education, training | High labor cost, low scalability | AI-based software, tools[58] | | √ |

*MEP* mechanical, electrical, and plumbing.

from design to operation. We assume that AI could help automate this process with much lower costs and minimal labor involvement. Table 2 shows some AI applications that can help buildings approach their theoretical maximum energy savings at a lower cost.

## AI reduces emissions of buildings

We believe that AI can improve energy efficiency and reduce carbon emissions through two main approaches: (1) AI helps scale up the best available technologies and practices. Because it can significantly help to scale up the technologies and speed adoption by reducing the construction and labor costs. Thus, it can lead to larger scale penetration of efficient technologies; (2) AI can further improve and optimize design, construction, and operation over the entire buildings' lifecycle, which brings in additional savings.

After we quantified the energy savings potential of individual HEEBs, the energy savings and emission reductions at scale needed to be extrapolated from our representative climate zones to the whole country. We developed six scenarios, including a Frozen scenario using the current building efficiency level as our baseline, two business-as-usual (BAU) scenarios with and without adopting AI, and three policy-driven scenarios promoting HEEBs and net-zero energy buildings (NZEBs) and even more aggressive policy implementation to

achieve zero emissions by 2050. The definitions of the six scenarios are listed in Supplementary Table 5.

All scenarios used the same building stock. In the Frozen scenario, the market shares of the baseline buildings (average energy efficiency), HEEBs, and NZEBs remain constant at the 2020 level throughout the future until 2050. However, in other scenarios, different market shares are employed to explore alternative scenarios and their impacts. The scenario with AI leads to a higher market share of HEEBs and NZEBs over time compared with the scenario without AI. This trend continues until the market share of net NZEBs reaches its maximum share.

The major policy measures include the promotion of efficiency technologies, implementation of building codes and energy efficiency standards, incentives, subsidies, financial assistance, and government-funded programs. These policies were constructed based on the pathway to the Long-Term Strategy of the United States[41], which aims for 100% clean electricity by 2035 and net-zero GHG emissions by 2050.

The policy scenario in our study encompasses four main pathways.
1. Decrease in the cost premium for highly energy-efficient (HEE) or net-zero energy (NZE) medium office buildings through investment on R&D and deployment of cost-effective technologies

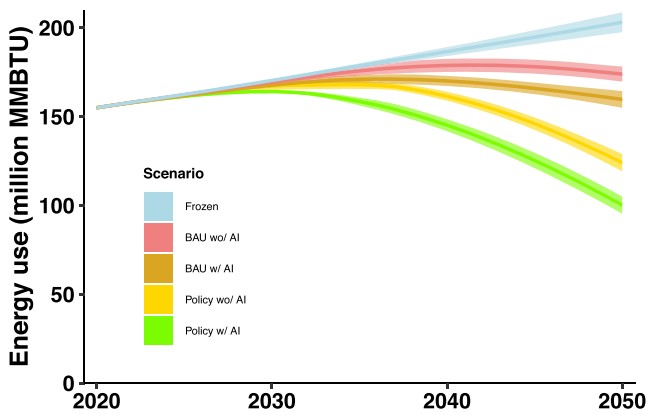

**Fig. 4 | Energy consumption by scenario.** BAU stands for business-as-usual. The darker color for each scenario indicates the average estimate and the lighter color for each scenario indicates the estimated ranges from the sensitivity analysis.

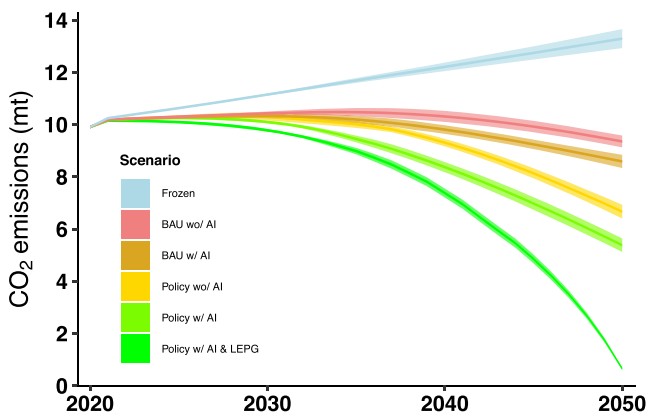

**Fig. 5 | $CO_2$ emissions by scenario.** BAU stands for business-as-usual; LEPG stands for low-emission power generation (decreasing from the 2020 level to zero emission by 2050). The darker color for each scenario indicates the average estimate and the lighter color for each scenario indicates the estimated range from the sensitivity analysis.

including additive manufacturing, incentives, subsidies, financial assistance, and government-funded programs.

2. Increase in the retrofit share of existing commercial buildings to improve their energy efficiency through building codes and energy efficiency standards, incentives, subsidies, financial assistance, and government-funded programs.

3. Increase in the share of new commercial buildings that achieve NZE status until the maximum allowed share is reached for each region or climate zone through investment on R&D and deployment of cost-effective technologies including additive manufacturing, building codes and energy efficiency standards, incentives, subsidies, financial assistance, and government-funded programs.

4. Increase in the share of retrofitted commercial buildings that achieve NZE status until the maximum allowed share is reached for each region or climate zone through building codes and energy efficiency standards, incentives, subsidies, financial assistance, and government-funded programs.

Sensitivity analysis was conducted to accommodate the uncertainty in the energy use projection. Based on uncertainty analysis and the pathway to the US Long-Term Strategy[41], the following assumptions were adopted for the sensitivity analysis: (1) the uncertainty will increase to ~10% by 2050 for the Frozen scenario; (2) the uncertainty

will increase to ~30–40% around 2030–2050 for the BAU scenarios; (3) the uncertainty will increase to ~20–30% around 2035 and then decrease to ~10% around 2050 for the Policy scenarios.

Figures 4 and 5 show the analysis results of the projected total final energy consumption and $CO_2$ emissions of the six scenarios, respectively. Supplementary Tables 6 and 7 summarize the analysis results of the projected total final energy consumption and $CO_2$ emissions of the six scenarios, respectively.

As shown in Fig. 4, if there is no energy efficiency improvement and policy support, building energy consumption will keep increasing, as described by the Frozen scenario. Considering continued technology improvement and increased shares of HEEBs and NZEBs in new construction and building retrofits, the BAU scenario without AI will peak around 2040. AI could help reduce the cost premium of HEE and/or NZE buildings and therefore could increase their market shares[23,24]. It was estimated that cost savings and revenue generation from adopting AI could be >10% of annual on-site energy costs[42], and AI, together with robotics and the Internet of Things, can reduce building costs by up to 20%[43]. In this study, we assumed that AI could help reduce building costs by 10%. Adopting AI will help the BAU scenario peak earlier, around 2035, and reduce energy consumption by ~8% in 2050 compared to BAU, or by ~21% compared to the Frozen scenario. Meanwhile, if energy efficiency policies (such as retrofit programs, incentives, rebates, or subsidies) are implemented, the cost premium of HEE/NZE buildings will be further reduced, which will further increase the share of NZEBs, even though it may be difficult and not cost-effective to reach this high share. Finally, to achieve the carbon neutrality target, the low-emission power generation (LEPG) was assumed to decrease carbon emission from the 2020 level to zero emissions by 2050. Adopting AI will help the policy scenario further reduce energy consumption by ~19% in 2050 compared to the policy scenario without AI. As a result, when integrated with energy policy and AI, the best scenario is expected to reach ~40 and 50% reductions in energy consumption in 2050 compared with the BAU and Frozen scenarios, respectively.

Similarly, Fig. 5 shows the analysis results of the $CO_2$ emissions. Adopting AI will help the BAU scenario peak earlier, around 2035, and reduce $CO_2$ emissions by ~8% in 2050 compared to the BAU scenario, or by ~35% compared to the Frozen scenario. Adopting AI will help the policy scenario further reduce $CO_2$ emissions by ~19% in 2050 compared to the policy scenario without AI. As a result, efficiency policy implementation along with AI technology could reduce $CO_2$ emissions by ~40% compared with the BAU scenario and 60% compared with the Frozen scenarios in 2050, respectively. Adding LEPG would help achieve a near zero-emission target in 2050, with ~93% savings from the BAU scenario and ~95% from the Frozen scenario, respectively.

## Discussion
### Limitations
Instead of concentrating on a specific AI technology, this paper utilizes engineering and energy simulation methods to quantify the potential impact of AI on enhancing energy efficiency and reducing carbon emissions in general. Enabled by advanced AI algorithms and techniques, data-driven modeling and decision-making approach can provide customized solutions and greatly enhance the adoption and implementation of high-energy technologies at scale with low cost. To further improve the accuracy and applicability of the proposed methodology, the applications of advanced control models (such as deep learning or reinforcement learning) can further be explored by following the same framework in future work.

While this paper takes medium office as an example, the proposed methodology can be applied to different commercial building types with appropriate adjustments of the input parameters based on specific characteristics and energy consumption patterns of each building category. To develop a comprehensive understanding of AI's energy

savings potential and carbon reduction opportunities across diverse building types, future work can expand the analysis to encompass a wider range of commercial and institutional buildings.

As AI continues to evolve rapidly, including recent advancements such as generative AI and large language models, there is potential for future research to track the evolving impact of AI beyond the scope of this study. Literature reviews indicate a wide range of diverse impacts across various domains and applications resulting from AI adoption, which highlights both significant potentials and uncertainties. Further investigations are required to explore this breadth and depth more comprehensively.

## AI's impact at scale

AI has been widely adopted in many application domains. In the building industry, it has been treated as an emerging technology that can be used to reduce energy consumption and carbon emissions to adapt to climate change. However, AI's saving potentials in buildings are not well understood and remain difficult to quantify. To fill the gaps, this paper:

- Proposes a four-key structure to systematical decouple and evaluate the theoretical maximum energy-saving potentials across an individual building's lifetime. The saving potentials vary based on different climate zones. AI is assumed to help buildings achieve these potentials at a lower cost.
- Quantifies how AI can help increase the adoption rate of low-energy use/NZEBs by developing energy efficiency technology adoption and building stock modeling. To predict the AI impact at scale, we considered the construction costs and deep retrofit costs of new and existing medium office buildings by climate zone and calculated the building stock turnover and technology adoption. The benefit of introducing AI technology was also analyzed.

This research shows that AI could help reduce the cost premium of HEEBs and NZEBs and therefore increase their market share penetrations. Adopting AI technology at scale is expected to decrease US medium office buildings' energy consumption and $CO_2$ emissions by ~8% compared with the BAU scenario without AI and ~19% compared to the policy scenario without AI in 2050, respectively. Integrating AI with energy efficiency policies and LEPG shows energy use decreasing by ~40% and $CO_2$ emissions by ~90% from the BAU scenario in 2050.

This research can help provide policymakers with quantitative decision support on energy saving and carbon reduction when promoting AI in the building industry. As a generic methodology, a similar approach can also be applied to estimate AI's savings potential for other building types and for other regions or countries. As discussed previously, by utilizing the same analytical framework and considering the unique characteristics of different building types, the proposed method can provide valuable insights and high-level conclusions applicable to a broader range of cases.

## Methods

### Modeling of individual building's energy-saving potentials

The discrepancy between the median US office and the median US zero energy verified building is assumed to be the technical building energy-saving potentials. We propose four key categories to account for this energy-saving potential, including imperfect design/construction, subpar controls/operation, occupancy influence, and equipment efficiency, as shown in Fig. 3. The total technical energy saving is described in Eq. (1).

$$S = S_1 + S_2 + S_3 + S_4 = (\alpha_1 + \alpha_2 + \alpha_3 + \alpha_4) \times EUI_{base} \quad (1)$$

where $S_1$-$S_4$ are the technical savings from equipment, occupancy influence, subpar controls/operation, and imperfect design/construction, respectively, and $\alpha_1$-$\alpha_4$ are the saving potentials (%).

To understand and quantify each $\alpha$ term in Eq. (1), this study conducted annual building energy simulations using the US DOE's EnergyPlus tool and collected some simulation results from literature review.

The ASHRAE standard 90.1 prototype building model was used for this study. The prototype building models[44] were developed by PNNL. They were derived from the DOE's Prototype Building Models with modifications based on input from the ASHRAE Standard 90.1 committee, the Advanced Energy Design Guide series, and building industry experts. These prototype buildings represent various building types and are well-calibrated based on real data. They have been used by the US DOE's Building Energy Codes Program for evaluating building energy codes and proposing code changes. A medium office was selected as an example building type in this study. Supplementary Fig. 1 shows the representative geometry and thermal zoning of the prototype medium office building defined by the ASHRAE standard 90.1. It has three floors, with a total floor area of 4980 square meters ($m^2$). Each floor has one core zone (60% of floor area) and four perimeter zones (40% of floor area). A packaged air conditioning unit was assumed to provide cooling and heating.

EnergyPlus building energy models were developed to consider different design variations caused by the proposed four key categories. Supplementary Tables 2 and 3 show the energy potentials from equipment and building design/construction, respectively.

### Energy efficiency technology adoption and building stock modeling

Consider the following discrete choices model[45,46]

$$\frac{s_{i,t}}{S_t} = \frac{a_{i,t}\exp(u_{i,t})}{\sum_{i=0}^{N-1} a_{i,t}\exp(u_{i,t})} \frac{s_{i,t}}{S_t} = \frac{a_{i,t}\exp(u_{i,t})}{\sum_{i=0}^{N-1} a_{i,t}\exp(u_{i,t})}, \quad (2)$$

where $i = 0,1,\cdots,N-1$ denotes the $i$th choice; $t = 0,1,\cdots,M-1$ denotes the $t$th time period; $s_{i,t}/S_t$ is the market share of choice $i$ for the specific end use in time period $t$, $\sum_{i=0}^{N-1} s_{i,t}/S_t = 1$; $a_{i,t}$ is the availability of choice $i$ for the specific end use in time period $t$; and $u_{i,t}$ is the utility of choice $i$ for the specific end use in time period $t$. For simplicity, we assume that index $i = 0$ denotes baseline building, $i = 1$ denotes HEEB, and $i = 2$ denotes NZEB.

We used net present value (NPV) to evaluate the utility of each type of building. Specifically, the NPV of adoption of a building type was calculated as the weighted sum of a projected stream of current and future benefits and costs[47]:

$$NPV = NB_0 + d_1 NB_1 + d_2 NB_2 + \cdots + d_n NB_n = \sum_{t=0}^{n} d_t NB_t, \quad (3)$$

where $t = 0,1,2,\cdots,n$ is time period index; $NB_t = B_t - C_t$ is the net difference between benefit ($B_t$) and cost ($C_t$) that accrue at the time period $t$; and $d_t$ is the discounting weight, with $d_0 = 1$ and $d_t = \frac{1}{(1+r)^t}$, where $r$ is the real discount rate.

The total conceptual first cost (TCC) considers only construction cost (CC) and development cost (DC), specifically,

$$TCC_j = CC_j + DC_j, \quad (4)$$

where: $j = 0,1,2$ denotes baseline building, HEEB, and NZEB, respectively. We further assumed that the costs of HEEB and NZEB are higher than baseline building, i.e.,

$$TCC_k = (CC_0 + \triangle CC_k) + (DC_0 + \triangle DC_k) = CC_0 \times (1+\mu_k) + DC_0 \times (1+\nu_k), \quad (5)$$

where: $k = 1,2$ denotes HEEB and NZEB, respectively; construction cost premium $\Delta CC_k = CC_0 \times \mu_k$ and development cost premium $\Delta DC_k = DC_0 \times \nu_k$, with construction cost premium percentage $\mu_k > 0$ and development cost premium percentage $\nu_k > 0$. The assumed values of $\mu_k$ and $\nu_k$ are listed in Supplementary Table 8.

The construction cost estimates used for each climate zone are listed in Supplementary Table 9 for new building and Supplementary Table 10 for energy retrofit of an existing building.

We assumed that cost premiums of the HEEB and NZEB would decline over time autonomously or due to policies:

$$TCC_{k,t} = [CC_0 + \triangle CC_k \times (1 - \alpha_{k,t})] + [DC_0 + \triangle DC_k \times (1 - \beta_{k,t})]$$
$$= CC_0 \times [1 + \mu_k \times (1 - \alpha_{k,t})] + DC_0 \times [1 + \nu_k \times (1 - \beta_{k,t})], \quad (6)$$

where $\alpha_{k,t}$ $(0 \leq \alpha_{k,t} \leq 1)$ denotes the construction cost premium decline percentage and $\beta_{k,t}$ $(0 \leq \beta_{k,t} \leq 1)$ denotes the development cost premium decline percentage, $t = 0,1, \cdots, T$.

As discussed previously in Sections 1 and 2, the utilization of AI could further reduce both the construction cost premium and the development cost premium, i.e.,

$$TCC_{k,t} = [CC_0 + \triangle CC_k \times (1 - \alpha_{k,t} - \xi_{k,t})] + [DC_0 + \triangle DC_k \times (1 - \beta_{k,t} - \zeta_{k,t})]$$
$$= CC_0 \times [1 + \mu_k \times (1 - \alpha_{k,t} - \xi_{k,t})] + DC_0 \times [1 + \nu_k \times (1 - \beta_{k,t} - \zeta_{k,t})], \quad (7)$$

where $\xi_{k,t}$ $(0 \leq \xi_{k,t} \leq 1)$ and $\zeta_{k,t}$ $(0 \leq \zeta_{k,t} \leq 1)$ are the contributions of AI to further decrease the construction cost premium and development cost premium on top of the autonomous or policy-induced declines over time, $t = 0,1, \cdots, T$. We further assumed that Eq. 7 is subject to the constraints $0 \leq 1 - \alpha_{k,t} - \xi_{k,t} \leq 1$ and $0 \leq 1 - \beta_{k,t} - \zeta_{k,t} \leq 1$.

**Assumptions.** This study adopted the following assumptions to calculate the NPV of each type of building:
- A 20-year time horizon with a discount rate of 20%[48].
- Operation and maintenance cost: $9.135 per sqft[49].
- Energy cost: $2.14 per sqft[50].
- Supplementary Tables 8 and 9 summarize the cost premium for HEEBs and NZEBs compared with the baseline buildings[51].
- Supplementary Table 11 summarizes the decrease in cost premium for HEEBs and NZEBs[51] due to the introduction of AI.
- Supplementary Table 12 shows the annual retrofit share of the total surviving medium office floor space.
- Considering the national solar energy distribution and the total rooftop areas of all medium office buildings[34,52], the shares of NZEBs cannot exceed the maximum allowed values as shown in Supplementary Table 13. Similarly, Supplementary Table 14 shows the maximum allowed NZEB share of the retrofitted medium office buildings.

## Data availability
The building energy consumption and carbon emission data generated in this study are provided in the Supplementary Information.

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

## Acknowledgements

This work was supported by Lawrence Berkeley National Laboratory (LBNL) under Contract No. DE-AC02-05CH11231. The United States Government retains, and by accepting the article for publication, the publisher acknowledges that the United States Government retains, a non-exclusive, paid-up, irrevocable, worldwide license to publish or reproduce the published form of this work, or allow others to do so, for United States Government purposes.

## Author contributions

C.D.: conceptualization, methodology, formal analysis, visualization, and writing/original draft. J.K.: methodology, formal analysis, visualization, and writing/original draft. M.L.: conceptualization, writing/reviewing, and editing. J.G.: conceptualization and visualization. N.Z.: conceptualization, supervision, writing/reviewing, and editing.

## Competing interests

The authors declare no competing interests.
