## [Peer Review File · Nature Communications]

Potential of Artificial Intelligence in Reducing Energy and Carbon Emissions of Commercial Buildings at ScaleReviewers' Comments:

Reviewer #1:

Remarks to the Author:

The article is well prepared. It covers aspects within the area of Jurnal. In my opinion no further changes are needed.

Reviewer #2:

Remarks to the Author:

This paper developed the AI-based method for reducing building energy consumption and emission. However, it cannot show great value in theory and practicability. It cannot meet the high requirement of this journal. The biggest problem is the novelty and the value of conclusions. It is hard to see the great value of AI in the targeted field. The comments are given below:

1. In fact, AI has been widely adopted in green building design. There are many successful applications with the help of various intelligent algorithms. Hence, it is improper to say that AI in improving energy efficiency and reducing carbon emissions has not been studied systematically. In this regard, the research gap raised by the author is doubtful. To compare your work with other existing ones, what are the merits of your work? What improvement have you made?
2. Can you provide the detail of the efficient policy implementation? This is the key content of your work. However, it is unclear. Other readers could be hard to follow and repeat your work. What is the policy? How can we obtain the policy and carry out it? Does it a data-driven policy making process? How can you test the effectiveness of the policy?
3. From the method part, the reviewer does not agree that this is an AI method. Instead, it can only be regarded as a mathematical modeling method. How can we see the intelligence from your method? Are there any parts with the function of prediction and optimization in your method? This paper is not innovative enough.
4. Is the method generalized enough to extend to other cases? This journal hopes the authors to provide some high-level conclusions, which cannot limit to a certain case. In the current version, it lacks some useful and convincing results.
5. Does the data used in this paper come from the EnergyPlus simulation tool? Is it reliable? The implementation of the tool will influence the quality of data. Do you have some real data to be used?

Reviewer #3:

Remarks to the Author:

Overall, this is a well written manuscript with an interesting approach to estimate potential for AI. That said, this paper does require a bit of work.

- Title could use a modification, as this discusses more about the potential of AI.
- The authors need to elaborate on each of their cases. For example, the authors explain that Case 1 increases the cooling efficiency by 20%. Case 2 increases the heating efficiency by 12%. Please justify how and why were such values selected? Moreover, how would the selection of percentages selected by the authors effect the overall analysis?
- Tables which summarize each energy evaluation would be beneficial to the readers.
- "We believe that AI can improve energy efficiency and reduce carbon emissions through two main approaches: (1) AI helps scale up the best available technologies and practices. Because, tt can significantly help to scale up the technologies and speed adoption by reducing the construction and labor costs." Any examples of this happening? Please provide a reference(s) if possible.

- How well would such an analysis apply for other advance control systems? AI is shown to potentially save certain percentages. However, model predictive control (using RC models for example) may also potentially have a similar savings. Would your analysis differ for different types of advance controls? A sentence or two describing how the methodology within could be applied to other advanced control techniques/technologies would be beneficial

There are many formatting issues as well

- Main body text sometimes appears to be in different size fonts
- Tables heading are on one page while the table is on a second (e.g. Table 1)
- Tables are shown overlapped onto two pages
- Are lines 160 to 164 a note? normal paragraph?
- Figure 2 can be difficult to read. Please improve the quality
- Line 134, Case 17?

Spelling and grammar need to be checked

Reviewer #4:

Remarks to the Author:

This study claims that AI can help reduce energy consumption and carbon emissions of buildings at scale and uses case studies for demonstration. This reviewer has the following comments for improvement.

1. The role of AI is vague. This reviewer did not see clearly the contributions of AI in this study. It seems that the simulation techniques are used to solve the problem in this study. However, AI is not the same to the simulation tool. What are the data-driven modeling and decisions?
2. The uncertainty in future predictions (e.g., 2050) of energy consumption and emission reduction is not considered in this research, which is not realistic in practices. This indicates that the results and findings are not reliable.
3. In Fig. 4 and Fig. 5, what does "Frozen" mean? More explanations are needed.
4. The scenarios with and without AI are not clearly presented.

Reviewer #5:

Remarks to the Author:

Key results

The manuscript deals with the role of artificial intelligence to improve energy efficiency and reduce carbon dioxide emissions in U.S. medium office buildings and aims to quantify potential reductions for different 2020-2050 scenarios. Alongside with energy policy and low carbon power generation, AI could help reduce up to energy use by 47% and GHG emissions by 93%.

Validity

The validity and robustness of the data interpretation and conclusions are in general appropriate. However, there are several flaws, which should be addressed. One of the main drawbacks is that the authors focus only on medium-sized office buildings to evaluate energy consumption and carbon dioxide reduction in U.S. buildings as stated in lines 19-20. Reasons for considering medium-sized office buildings should be provided, along with suitable statistics to support this approach such as the market share of medium-sized office buildings as well as their contribution to the total building energy use and/or GHG emissions in buildings.

Another important flaw is how AI can be applied to improve building performance. Several applications are provided, including equipment, imperfect design, imperfect construction, subpar controls and operation. In some of the proposed examples, AI could be helpful but it might not be required. For instance, a rule-based FDD with no AI could possibly provide similar savings compared to an AI-based FDD whereas MPC could rely on white-box or grey-box models, rather than AI models, while achieving similar energy savings. In this specific case, the AI term encompasses improved controls and operation in general, not just AI-based controls and operation, which could be a bit misleading about the potential of AI. The wording could be modified accordingly. Similar remarks could be done for the other applications.

Finally, the study focusses on buildings with packaged air conditioning units equipped with natural gas furnace and the authors evaluated potential increases in cooling and heating efficiency in Supplementary Table 2. However, alternatives to such a conventional system have not been investigated while some options could significantly affect study results. In the context of 2050 emission targets, heat pumps might offer more potential for energy and cost savings, might be more suitable especially for HEE and NZE buildings while possibly allowing carbon neutral buildings by 2050.

Significance

The manuscript provides reasonable estimates of the potential of AI for supporting the decarbonization of buildings to meet 2050 emission target. Such a piece of work helps promote the need for advanced tools targeting improved design, construction, operation and maintenance by assessing associated savings, and could support the development of new policy promoting the use of AI tools and data-driven solutions in general.

Data and methodology, analytical approach

The proposed methodology to evaluate how AI could help reduce energy consumption and carbon emissions in buildings is interesting and seems correct. Nonetheless, some clarifications are required to better understand the approach. Firstly, the method to estimate potential savings related to AI is confusing. In the paragraph at lines 66-72, the energy use intensity is provided for median U.S. office (167 kWh/m²) and for low energy verified buildings (57 kWh/m²). However, it is not clear how this value of 57 kWh/m² is used afterwards in the manuscript. Indeed, the proposed approach intends to estimate savings from a baseline (the median U.S. office) and might not need to refer to the low energy verified building EUI. In addition, Figure 3 also shows the median of zero (not low) energy verified building in the U.S.

The "policy w/AI & LEPC" allows to reach up to 95% emission savings. More information could be provided to describe how this low emission power generation can be achieved and why buildings cannot be carbon neutral (my guess is the natural gas furnace in packaged air conditioners).

The manuscript lacks information regarding the extrapolation from individual buildings to the whole country. The method should be presented along with the main assumptions (share of buildings per ASHRAE climate zone, etc.) and associated references.

The impact of AI algorithms is modelled with a factor to decrease in energy usage and cost in Eq. 7. This decrease might be put in perspective with side effects such as the CPU requirements and capacities resulting in increased electric power for the deployment of AI at large scale. I do not necessarily expect precise numbers but rather a discussion on the implication of using AI at large scale.

Lastly, results given in Figures 4 and 5 show the impact of AI along with other measures such as new energy policy and low carbon power generation. Providing the individual impact of AI in % in the main text could be of interest with regards to the scope of the manuscript.

Suggested improvements

I suggest to include the option to replace packaged air conditioners by heat pump units, especially for HEE and NZE buildings. Another important element could be the extension of the present study to commercial and institutional buildings at large, not only medium sized office buildings; nonetheless, this aspect is optional since it might require significant changes. However, the authors could provide more information about the methodology to come up with such an analysis.

Clarity and context

The manuscript is well written and easy to follow. A short description of the paper organization could be provided after line 62 to help the reader better follow the approach. Some typos were found throughout the text and should be corrected. The y axis label in Figure 2 shows "energy", not "energy"; the y axis label in Figure 2 – right should be modified to "Energy savings from building design and construction". In Table 2, there is an extra row in "operation", which should be removed. At line 170, "tt" should be corrected to "it". "Subsides" at line 197 might be corrected by "subsidies". In Supplementary Table 11, 2021 is given instead of 2020.

References

Overall, the manuscript references previous literature appropriately. However, additional references could be provided to support the occupancy influence energy savings of 15-20% (line 147) and the information provided in Table 2 (AI application examples with energy and cost saving potential). Numbers given in lines 306-320 could also be broken down for each assumption, instead of being lumped together in the line 307. Additional work could be performed to break down long lists of references (e.g. 10-18 line 46; 19-33 line 50; 26,28,29,38,39 line 193; 24-26,28-29,34,39,43-47 line 307) whereas references could be added to support the method section (no references are provided).

Etienne Saloux (Natural Resources Canada, CanmetENERGY)

Response to Reviewers

We would like to express our sincere gratitude to the reviewers for your valuable comments and suggestions. Your feedback has greatly contributed to the improvement of the manuscript. We have carefully considered each comment and made necessary modifications to address the raised concerns. All the revisions have been clearly highlighted in the revised manuscript for easy reference. Please refer to our point-to-point responses below for a detailed overview of the changes implemented.

Reviewer #1 (Remarks to the Author):

The article is well prepared. It covers aspects within the area of Journal. In my opinion no further changes are needed.

Authors' response:

Thank you for your positive feedback and for considering the article well-prepared. We appreciate your assessment that the article covers relevant aspects within the scope of the journal. Your opinion that no further changes are needed is valuable and reassuring. We will address any remaining comments or suggestions from the other reviewers and make necessary revisions accordingly. Your feedback is greatly appreciated.

Reviewer #2 (Remarks to the Author):

This paper developed the AI-based method for reducing building energy consumption and emission. However, it cannot show great value in theory and practicability. It cannot meet the high requirement of this journal. The biggest problem is the novelty and the value of conclusions. It is hard to see the great value of AI in the targeted field. The comments are given below:

Question1:

1. In fact, AI has been widely adopted in green building design. There are many successful applications with the help of various intelligent algorithms. Hence, it is improper to say that AI in improving energy efficiency and reducing carbon emissions has not been studied systematically. In this regard, the research gap raised by the author is doubtful. To compare your work with other existing ones, what are the merits of your work? What improvement have you made?

Authors' response:

Thank you for your valuable feedback. We acknowledge that there have been successful applications of AI in green building design. However, our intention was to highlight the lack of studies that comprehensively assess the overall impact of AI on energy efficiency and carbon emissions reduction. While individual case studies exist, their focus is often limited to specific aspects of building performance such as design, construction, or operation. Consequently, these studies often yield a wide range of potential energy savings, ranging from 2% to 60%. This wide variation can be less informative and challenging for decision-makers seeking clearer and more actionable insights.

The novelty of our work lies in two key aspects. Firstly, we provide a holistic perspective by considering all four key categories: equipment, occupancy influence, control and operation, and design and construction. This comprehensive approach sets our research apart from previous studies. Secondly, we analyze the impact of AI on a national scale using technology adoption and building stock modeling. Our findings indicate that AI can significantly contribute to energy consumption and carbon emissions reduction by reducing the cost of technology adoption at scale.

We realized that we might have not clearly articulate these novelties in our original submission. We appreciate your feedback, which has prompted us to emphasize the unique contributions of our research.

Question2:

2. Can you provide the detail of the efficient policy implementation? This is the key content of your work. However, it is unclear. Other readers could be hard to follow and repeat your work. What is the policy? How can we obtain the policy and carry out it? Does it a data-driven policy making process? How can you test the effectiveness of the policy?

Authors' response:

Thank you for raising important questions regarding the details of our efficient policy implementation. We appreciate the opportunity to provide further clarification.

The major policy measures employed in our scenario include the investment on R&D, deployment of cost-effective technologies including additive manufacturing, promotion of efficiency technologies, implementation of building codes and energy efficiency standards, incentives, subsidies, financial assistance, and government-funded programs. These policies are consistent with the pathway to the Long-Term Strategy of the United States, which aims for 100% clean electricity by 2035 and net-zero greenhouse gas emissions by 2050.

The policy scenario in our study encompasses four main pathways:

1. Decrease in the cost premium for highly energy-efficient (HEE) or net-zero energy (NZE) medium office buildings through investment on R&D and deployment of cost-effective technologies including additive manufacturing, incentives, subsidies, financial assistance, and government-funded programs.
2. Increase in the retrofit share of existing commercial buildings to improve their energy efficiency through building codes and energy efficiency standards, incentives, subsidies, financial assistance, and government-funded programs.
3. Increase in the share of new commercial buildings that achieve net-zero energy status until the maximum allowed share is reached for each region or climate zone through investment on R&D and deployment of cost-effective technologies including additive manufacturing, building codes and energy efficiency standards, incentives, subsidies, financial assistance, and government-funded programs.
4. Increase in the share of retrofitted commercial buildings that achieve net-zero energy status until the maximum allowed share is reached for each region or climate zone through building codes and energy efficiency standards, incentives, subsidies, financial assistance, and government-funded programs.

Specific details regarding these pathways can be found in Supplementary Table 9-12 of our manuscript.

While our policy scenario is based on relatively conservative assumptions, taking into account the timeline and economically justified maximum allowed shares for new and existing buildings, we also considered variations among different regions and the differences between residential and commercial applications. This approach allows us to generalize our results to the entire commercial buildings sector.

Although our policy-making process involves setting assumptions and pathways based on factors such as construction costs and renewable energy potential data, we acknowledge that it is not strictly a data-driven policy-making process. Instead, our policies are aligned with national and subnational policies already adopted such as in California and New York.

We appreciate your feedback and have addressed these points in our revised manuscript to provide a clearer understanding of the efficient policy implementation in our study. Please refer to paragraphs 3-5 in Section 3.

Question3:

3. From the method part, the reviewer does not agree that this is an AI method. Instead, it can only be regarded as a mathematical modeling method. How can we see the intelligence from

your method? Are there any parts with the function of prediction and optimization in your method? This paper is not innovative enough.

Authors' response:

Thank you for your valuable feedback.

We would like to clarify that the main objective of our study is not to focus on a specific AI method, but rather to generally explore the potential contribution of AI in enhancing energy efficiency and reducing carbon emissions. Our approach primarily utilizes mathematical modeling and engineering analysis to investigate how AI can help reduce construction time and costs, leading to the increased adoption of high-performance buildings. Our focus is on the broader potential of AI in driving improvements in energy efficiency and carbon reduction through economic and market effects.

Having said this, our methodology does leverage AI-related techniques indirectly. By considering various factors such as occupancy influence, control and operation, and design and construction, we incorporate predictive and optimization aspects in our analysis. These elements contribute to the overall goal of reducing energy consumption and carbon emissions by identifying cost-effective technology adoption strategies.

We appreciate your feedback regarding the level of innovation in our paper. While our study may not introduce entirely new AI methods, it provides valuable insights into the potential benefits of AI in achieving energy efficiency and carbon emissions reduction at scale. We will make sure to clarify these points and highlight the innovative aspects in our revised manuscript.

Question4:

4. Is the method generalized enough to extend to other cases? This journal hopes the authors to provide some high-level conclusions, which cannot limit to a certain case. In the current version, it lacks some useful and convincing results.

Authors' response:

Thank you for your feedback. We recognize the importance of a generalized method that can be applied to various cases. In our study, we utilized well-established engineering analysis techniques such as building energy simulation, technology adoption, and building stock modeling. These methodologies are widely accepted and commonly used in the US DOE's standard and regulation development processes.

Although we focused on a medium office building as an example in this paper, our methodology is not limited to this specific case. It can be extended and applied to other building types as well. By utilizing the same analytical framework and considering the unique characteristics of different building types, our method can provide valuable insights and high-level conclusions applicable to a broader range of cases.

Thank you for pointing out that we did not explicitly state the generalizability of our method in the current version. Your feedback has prompted us to clarify this aspect in our revised manuscript, ensuring that our conclusions are useful and convincing beyond the specific case presented.

Question5:

5. Does the data used in this paper come from the EnergyPlus simulation tool? Is it reliable? The implementation of the tool will influence the quality of data. Do you have some real data to be used?

Authors' response:

Thank you for your question regarding the reliability of the data used in our paper. The energy savings potential analysis is based on EnergyPlus simulations, which is a widely accepted and validated tool developed by the U.S. Department of Energy. EnergyPlus has been extensively used by researchers, practitioners, and regulatory agencies for building energy performance analysis.

In our study, we utilized prototype medium office buildings as examples, which are derived from DOE's Prototype Building Models and developed by the Pacific Northwest National Laboratory (PNNL). These prototype buildings (defined by ASHRAE standard 90.1) represent various building types and are well-calibrated based on real data. They have been used by the U.S. Department of Energy's Building Energy Codes Program for evaluating building energy codes and proposing code changes.

While we primarily relied on simulated data from EnergyPlus for our analysis, it is important to note that these simulations are based on validated models and real-world building characteristics. This provides a robust foundation for our findings.

Reviewer #3 (Remarks to the Author):

Overall, this is a well written manuscript with an interesting approach to estimate potential for AI. That said, this paper does require a bit of work.

Authors' response:

We appreciate your positive feedback on the manuscript and your recognition of our interesting approach to estimating the potential for AI. We have carefully reviewed your comments and suggestions, and we have made revision to address them. Please find our point-to-point response below, which outlines the specific changes we have made to improve the paper:

Question1:

- Title could use a modification, as this discusses more about the potential of AI.

Authors' response:

We appreciate your feedback on the title of our manuscript. Based on your suggestion, we have revised the title to better reflect the content and focus of the paper. The updated title is "Exploring the Potential of Artificial Intelligence in Reducing Energy Consumption and Carbon Emissions of Commercial Buildings at Scale". We believe that the new title accurately represents the discussion on the potential of AI. Thank you for bringing this to our attention, and we hope the revised title aligns more effectively with the scope of the paper.

Question2:

- The authors need to elaborate on each of their cases. For example, the authors explain that Case 1 increases the cooling efficiency by 20%. Case 2 increases the heating efficiency by 12%. Please justify how and why were such values selected? Moreover, how would the selection of percentages selected by the authors effect the overall analysis?

Authors' response:

Thank you for your valuable feedback and pointing out we did not provide sufficient details regarding the selection of percentages in each case. The values were chosen based on the difference of the energy performance of actual products and commercially available technologies.

To note, our analysis does not include the potential benefits of non-commercialized best available technologies, as the time needed for validation and commercialization is uncertain. We acknowledge that considering the non-commercialized technologies would likely result in higher energy saving potentials than our current estimates. Sensitivity analysis has been added in the later section to reflect the uncertainty from the energy savings potential.

We appreciate your input and have made the necessary revisions to address this concern. Thank you for bringing this to our attention.

Question3:

- Tables which summarize each energy evaluation would be beneficial to the readers.

Authors' response:

Thank you for your suggestion regarding the inclusion of tables summarizing each energy evaluation. We agree that tables would provide a clearer overview of the results and enhance the reader's understanding. In response to your comment, we have added relevant tables (Table 5-7) in the Supplementary Information document, which provide a comprehensive summary of the energy evaluations performed in our study. These tables include the key findings and metrics for each case analyzed. We believe that the addition of these tables will greatly benefit the readers in comprehending the results. Thank you for bringing this to our attention, and we appreciate your valuable input.

Question4:

- “We believe that AI can improve energy efficiency and reduce carbon emissions through two main approaches: (1) AI helps scale up the best available technologies and practices. Because, it can significantly help to scale up the technologies and speed adoption by reducing the construction and labor costs.” Any examples of this happening? Please provide a reference(s) if possible.

Authors' response:

Artificial intelligence (AI) has gained significant attention and is being increasingly applied in various industries, including the construction industry. Its potential to reduce construction and

labor costs, mitigate risks for construction workers, and enhance their health benefits has been recognized. Construction companies are increasingly realizing the advantages of utilizing AI in different aspects of construction, ranging from design to the implementation of cutting-edge technologies and practices.

Due to the relatively recent emergence of AI applications in the construction industry, the existing literature in this field is still developing. A technical review article titled "Artificial intelligence in the construction industry: A review of present status, opportunities, and future challenges" (Abioye et al., 2021. J Build Eng 44, 103299) provides valuable insights into the subject. References 21-23 in the manuscript also provide some good information on this topic. In addition to technical resources, there are non-technical articles that discuss the transformative impact of AI in the construction industry. "How AI is transforming the construction industry" (<https://theconstructor.org/artificial-intelligence/how-ai-is-transforming-the-construction-industry/568678/>) and "Who says you can't teach an old dog new tricks? The case for AI in construction" (<https://www.forbes.com/sites/angelicakrystledonati/2023/03/07/who-says-you-cant-teach-an-old-dog-new-tricks-the-case-for-ai-in-construction/?sh=79bf45fd3654>) are two interesting reads that highlight the potential benefits and opportunities presented by AI. While the literature on AI applications in the construction industry is still evolving, these resources provide valuable insights into the promising potential of AI in reducing costs, improving construction processes, and enhancing worker well-being.

Question5:

- How well would such an analysis apply for other advance control systems? AI is shown to potentially save certain percentages. However, model predictive control (using RC models for example) may also potentially have a similar savings. Would your analysis differ for different types of advance controls? A sentence or two describing how the methodology within could be applied to other advanced control techniques/technologies would be beneficial

Authors' response:

Thank you for your insightful question. Our analysis primarily focuses on currently widely adopted control measures for energy savings in buildings. We acknowledge that other advanced control systems, such as Reinforcement Learning or Deep Learning models, may have the potential for additional savings beyond what is captured in our analysis.

The methodology presented in our study can serve as a foundation for evaluating the energy savings potential of different advanced control techniques and technologies. By incorporating the specific characteristics and benefits of these advanced controls into the analysis framework, it is possible to assess their impact on energy efficiency and carbon emissions reduction.

In future work, we plan to explore the application of these advanced control methods and technologies, such as RC models or deep learning, within the context of our analysis. This will allow us to provide a more comprehensive understanding of their potential contributions to energy savings and further improve the accuracy and applicability of our methodology.

We appreciate your suggestion and added a “Discussions” section on the application of the methodology to other advanced control techniques in the revised manuscript. Now the Discussions section reads as below:

Instead of concentrating on a specific AI technology, this paper utilizes engineering and energy simulation methods to quantify the potential impact of AI on enhancing energy efficiency and reducing carbon emissions in general. Enabled by advanced AI algorithms and techniques, data-driven modeling and decision-making approach can provide customized solutions and greatly enhance the adoption and implementation of high energy technologies at scale with low cost. To further improve the accuracy and applicability of the proposed methodology, the applications of advanced control models (such as deep learning or reinforcement learning) can further be explored by following the same framework in future work.

While this paper takes medium office as an example, the proposed methodology can be applied to different commercial building types with appropriate adjustments of the input parameters based on specific characteristics and energy consumption patterns of each building category. To develop a comprehensive understanding of AI’s energy savings potential and carbon reduction opportunities across diverse building types, future work can expand the analysis to encompass a wider range of commercial and institutional buildings.

Question6:

There are many formatting issues as well

- Main body text sometimes appears to be in different size fonts
- Tables heading are on one page while the table is on a second (e.g. Table 1)
- Tables are shown overlapped onto two pages
- Are lines 160 to 164 a note? normal paragraph?
- Figure 2 can be difficult to read. Please improve the quality
- Line 134, Case 17?

Spelling and grammar need to be checked

Authors' response:

Thank you for bringing the formatting issues to our attention. We apologize for the errors and appreciate your feedback. In response to your comments, we have carefully reviewed and revised the manuscript to address the formatting issues you mentioned. We have ensured consistent font sizes throughout the main body text, adjusted the placement of table headings to align with their respective tables, and resolved any overlapping issues with the tables. We have also clarified the formatting of lines 160 to 164. They are not normal paragraph, but notes in the Figure following Nature Communications' format. Additionally, we have improved the quality of Figure 2 to enhance its readability. Furthermore, we have corrected the reference to "Case 17" in line 134. It should be "Case 9". Lastly, we have conducted a thorough spell and grammar check to ensure the manuscript is free of any errors. We appreciate your attention to detail and thank you for helping us improve the quality of our manuscript.

Reviewer #4 (Remarks to the Author):

This study claims that AI can help reduce energy consumption and carbon emissions of buildings at scale and uses case studies for demonstration. This reviewer has the following comments for improvement.

Question1:

1. The role of AI is vague. This reviewer did not see clearly the contributions of AI in this study. It seems that the simulation techniques are used to solve the problem in this study. However, AI is not the same to the simulation tool. What are the data-driven modeling and decisions?

Authors' response:

Thank you for bringing up this point. We realized that we might have not clearly explain the role of AI in our study. We thus would like to clarify that the objective of this study was not exploring specific AI technologies in depth. Instead, our main objective is to investigate the potential of AI in increasing the market shares of high energy efficiency and net zero energy buildings. We believe that AI has the capability to accelerate construction processes, reduce costs, and enhance safety and health benefits for construction workers. Our study aims to shed light on how AI can facilitate these improvements in the construction industry.

The use of simulation techniques, such as energy modeling and analysis, is indeed an important aspect of our study. However, we also recognize the potential for AI to play a significant role in data-driven modeling and decision-making processes related to energy efficiency and carbon emissions reduction in buildings.

In our analysis, we primarily utilize engineering and energy simulation methods to estimate the energy savings potential. However, we believe that data-driven modeling and decision-making, enabled by AI techniques, can greatly enhance the adoption and implementation of energy-efficient technologies at scale. By leveraging AI algorithms, such as machine learning and optimization, it is possible to optimize building operations, predict energy consumption patterns, and make data-driven decisions for energy efficiency measures.

We appreciate your comment and will make sure to provide a clearer explanation of the potential role of AI in data-driven modeling and decision-making in the revised manuscript.

Question2:

2. The uncertainty in future predictions (e.g., 2050) of energy consumption and emission reduction is not considered in this research, which is not realistic in practices. This indicates that the results and findings are not reliable.

Authors' response:

Thank you for your valuable feedback. While our main focus is on evaluating the potential contribution of AI in scaling up best practice technologies and practices, we acknowledge the importance of addressing uncertainties in our research. In response to your suggestion, we have conducted a sensitivity analysis to provide a better understanding of the uncertainties involved in our future predictions. This analysis helps to enhance the robustness and reliability of our results and findings.

Question3:

3. In Fig. 4 and Fig. 5, what does “Frozen” mean? More explanations are needed.

Authors’ response:

Thank you for your valuable feedback. In our study, the term "Frozen" refers to a scenario where the market shares of three types of buildings (baseline buildings, high energy efficiency buildings, and net zero energy buildings) remain constant at the 2020 level throughout the future until 2050. This scenario serves as a counterfactual baseline for comparison in our analysis. We will make sure to provide a more detailed explanation in the figure captions to avoid any confusion.

Question4:

4. The scenarios with and without AI are not clearly presented.

Authors’ response:

Thank you for your valuable feedback. We added a new Supplementary Table 5 to explain the definitions of each scenarios. All the pathways related to the scenarios are listed in the supplementary Tables 11-14. In our study, the key distinction between the scenarios lies in the role of AI in accelerating the design, implementation, and operation of energy efficiency technologies throughout the entire building lifecycle. Additionally, the scenarios with AI incorporate an additional 10% reduction in construction and labor costs compared to the scenarios without AI. This cost reduction enhances the competitiveness and subsequent adoption of energy efficiency technologies throughout the building lifecycle.

To elaborate on the implementation of AI in our study, we utilized discrete choice modeling to capture the impact of AI on cost reduction, which, in turn, drives the increased adoption of high energy performance (low EUI) buildings and net zero energy buildings. With the inclusion of AI, the adoption rate of high energy efficiency buildings and net zero energy buildings is higher due to the reduction in construction time and costs.

In summary, the scenario with AI leads to a higher market share of high energy efficiency buildings and net zero energy buildings over time compared to the scenario without AI. This trend continues until the market share of net zero energy buildings reaches its maximum share.

Reviewer #5 (Remarks to the Author):

Thank you for providing such detailed comments on our paper. Your feedback has been immensely valuable and has greatly contributed to the improvement of the paper's quality. We truly appreciate the effort and thoughtfulness you put into reviewing our work.

Your insightful comments have inspired us to make significant revisions to address the specific points you raised. We carefully considered your suggestions and incorporated them into the revised manuscript. Your input has undoubtedly strengthened the overall content and clarity of our paper.

Once again, we extend our sincere gratitude for your thorough review and valuable insights. Your contribution has been instrumental in enhancing the quality of our work. Please find our point-to-point response below for your review.

Key results

The manuscript deals with the role of artificial intelligence to improve energy efficiency and reduce carbon dioxide emissions in U.S. medium office buildings and aims to quantify potential reductions for different 2020-2050 scenarios. Alongside with energy policy and low carbon power generation, AI could help reduce up to energy use by 47% and GHG emissions by 93%.

Validity

The validity and robustness of the data interpretation and conclusions are in general appropriate. However, there are several flaws, which should be addressed.

Question 1:

One of the main drawbacks is that the authors focus only on medium-sized office buildings to evaluate energy consumption and carbon dioxide reduction in U.S. buildings as stated in lines 19-20. Reasons for considering medium-sized office buildings should be provided, along with suitable statistics to support this approach such as the market share of medium-sized office buildings as well as their contribution to the total building energy use and/or GHG emissions in buildings.

Authors' response:

Thank you for your comment. Although our paper aims to establish a methodology that can be widely applied to assess AI impact to building energy consumption reductions, we acknowledge the importance of providing a clear rationale for focusing on medium-sized office buildings in our study. We agree that it would be beneficial to include suitable statistics to support this approach and highlight the significance of medium office buildings in the context of total building energy use and greenhouse gas emissions.

In our revised manuscript, we will include the statistics to show the importance of medium office. It dominates the energy consumption of all office buildings.

Thank you for bringing this to our attention, and we appreciate your feedback in helping us strengthen our study.

Question 2:

Another important flaw is how AI can be applied to improve building performance. Several applications are provided, including equipment, imperfect design, imperfect construction, subpar controls and operation. In some of the proposed examples, AI could be helpful but it might not be required. For instance, a rule-based FDD with no AI could possibly provide similar savings compared to an AI-based FDD whereas MPC could rely on white-box or grey-box models, rather than AI models, while achieving similar energy savings. In this specific case, the AI term encompasses improved controls and operation in general, not just AI-based controls and operation, which could be a bit misleading about the potential of AI. The wording could be modified accordingly. Similar remarks could be done for the other applications.

Authors' response:

Thank you for your insightful comment. We appreciate your perspective on the potential application of AI in improving building performance and the distinction between AI-based approaches and other techniques such as rule-based FDD and model predictive control.

In our revised manuscript, we will clarify the definition of AI and emphasize that our focus is on the potential of AI to enhance the adoption of energy-saving measures at a large scale, rather than analyzing one specific AI approach. We agree that there are various techniques, including rule-based methods and model-based control, that can contribute to energy savings without explicitly relying on AI models. We believe that AI could help speed up the adoption of energy efficiency technologies and advance the development of high-performance buildings by reducing construction time and costs, while also enhancing the health and well-being of construction workers.

By modifying the wording accordingly, we aim to provide a more accurate representation of the role of AI and its potential in improving building performance, while acknowledging the effectiveness of other approaches. We appreciate your valuable input in helping us refine our paper.

Question 3:

Finally, the study focusses on buildings with packaged air conditioning units equipped with natural gas furnace and the authors evaluated potential increases in cooling and heating efficiency in Supplementary Table 2. However, alternatives to such a conventional system have not been investigated while some options could significantly affect study results. In the context of 2050 emission targets, heat pumps might offer more potential for energy and cost savings, might be more suitable especially for HEE and NZE buildings while possibly allowing carbon neutral buildings by 2050.

Authors' response:

Thank you for your valuable comment. We agree that alternative systems, such as heat pumps, have the potential to significantly impact energy and cost savings, particularly in the context of

meeting 2050 emission targets and achieving carbon-neutral buildings. We appreciate your suggestion to include heat pump options in our analysis.

In the revised manuscript, we have included one new scenario (Case 9) under the Equipment efficiency category specifically addressing heat pump solutions. It considers the replacement of packaged air conditioning units with heat pumps for space heating. We have modeled and estimated the technical energy savings associated with this scenario.

By incorporating these heat pump scenarios, we aim to provide a more comprehensive analysis of potential energy savings and further explore the role of heat pumps in achieving energy efficiency and carbon reduction goals. We thank you for bringing this important aspect to our attention and contributing to the enhancement of our study.

To ensure the generalizability of our conclusions beyond medium office buildings, we considered the challenges faced by large commercial buildings, particularly in cold or very cold climate zones that require significant heating capacity within short time frames. Given the limited availability and higher costs of current heat pump systems for such requirements, we included natural gas heating as a baseline in our study. However, we recognize that advancements in heat pump technologies may make them more competitive in the future. In our future research, we plan to investigate the maximum technical and economic potential of heat pump systems.

For this study, we took a conservative approach by adopting realistic assumptions and pathways that consider the limitations of existing heat pump products and the substantial heating needs of certain large commercial buildings. Our primary objective is to assess the contribution of AI in achieving energy savings and reducing carbon emissions in commercial buildings by promoting the adoption of high energy efficiency and net zero buildings. We believe that our methodology, with its conservative assumptions, can be applied to various types of commercial buildings with appropriate adjustments, allowing for the potential impact of AI to be evaluated across the sector.

Significance

The manuscript provides reasonable estimates of the potential of AI for supporting the decarbonization of buildings to meet 2050 emission target. Such a piece of work helps promote the need for advanced tools targeting improved design, construction, operation and maintenance by assessing associated savings, and could support the development of new policy promoting the use of AI tools and data-driven solutions in general.

Question 4:

Data and methodology, analytical approach

The proposed methodology to evaluate how AI could help reduce energy consumption and carbon emissions in buildings is interesting and seems correct. Nonetheless, some clarifications are required to better understand the approach. Firstly, the method to estimate potential savings related to AI is confusing. In the paragraph at lines 66-72, the energy use intensity is provided for median U.S. office (167 kWh/m²) and for low energy verified buildings (57 kWh/m²). However, it is not clear how this value of 57 kWh/m² is used afterwards in the manuscript. Indeed, the proposed approach intends to estimate savings from a baseline (the median U.S. office) and might not need to refer to the low energy verified building EUI. In addition, Figure 3 also shows the median of zero (not low) energy verified building in the U.S.

Authors' response:

Thank you for your valuable feedback. The primary objective of this study is not to investigate specific AI technologies. Instead, we aim to explore how AI can contribute to reducing energy consumption and carbon emissions by decreasing overall construction time and costs. More specifically, we investigate two aspects: (1) the acceleration of design, implementation, and operation and maintenance (OM) of energy efficiency technologies throughout the building lifecycle, and (2) the reduction of construction and labor costs. These outcomes would enhance the competitiveness and adoption of energy efficiency technologies in the overall building lifecycle. To achieve this, we employed discrete choice modeling in our study. This approach enables us to incorporate the cost reduction resulting from AI applications, which in turn amplifies the competitiveness and adoption of high energy efficiency (low EUI) buildings and net zero energy buildings.

Regarding the baseline EUI of low EUI buildings, it is set at 57 kWh/m². The EUI serves as the baseline for calculating energy savings when comparing low EUI buildings to high EUI buildings. For example, the unit energy savings achieved by adopting a low EUI building is $167 - 57 = 110$ kWh/m².

The “policy w/AI & LEFG” allows to reach up to 95% emission savings. More information could be provided to describe how this low emission power generation can be achieved and why buildings cannot be carbon neutral (my guess is the natural gas furnace in packaged air conditioners).

Authors' response:

We appreciate your comments regarding the low emission power generation and the use of natural gas in certain areas. In our study, we assume a transition towards 100% clean electricity generation as part of our low emission power generation scenario.

Meanwhile, we have taken into consideration the unique heating requirements of large-scale commercial buildings in cold and very cold areas. These types of buildings often have high heating capacity demands within short timeframes, and currently, the widespread use of heat

pumps for such applications is still challenging. The heating energy usage patterns of these large-scale commercial buildings differ significantly from residential buildings and small-scale commercial buildings.

Considering these factors, we have accounted for a small share of natural gas use in our analysis. This is to reflect the current competitiveness of natural gas in large-scale commercial buildings, especially in the near to medium terms. We believe that natural gas will continue to play a role in meeting the heating demands of these specific building types.

We appreciate your input, and we will ensure that these points are clarified and further elaborated upon in the revised version of the manuscript.

The manuscript lacks information regarding the extrapolation from individual buildings to the whole country. The method should be presented along with the main assumptions (share of buildings per ASHRAE climate zone, etc.) and associated references.

Authors' response:

Our study uses DOE's prototype building models in each climate zone to capture typical energy usage patterns. The prototype building models are developed by the Pacific Northwest National Laboratory (PNNL). These prototype buildings (defined by ASHRAE standard 90.1) represent various building types and are well-calibrated based on real data. They have been used by the U.S. Department of Energy's Building Energy Codes Program for evaluating building energy codes and proposing code changes. While we focused on medium office buildings, our model incorporates representative and cost-effective technologies applicable to various commercial building types. As a result, we believe our findings can be generalized to most building types, given our relatively conservative assumptions and pathways.

The impact of AI algorithms is modelled with a factor to decrease in energy usage and cost in Eq. 7. This decrease might be put in perspective with side effects such as the CPU requirements and capacities resulting in increased electric power for the deployment of AI at large scale. I do not necessarily expect precise numbers but rather a discussion on the implication of using AI at large scale.

Authors' response:

Thank you for your valuable suggestion. We would like to note that our study focuses on the energy impact of the AI application, not generic impact of AI technologies. We do acknowledge that the specific hardware and infrastructure requirements for the large-scale deployment of AI should be an important topic for future studies.

Lastly, results given in Figures 4 and 5 show the impact of AI along with other measures such as new energy policy and low carbon power generation. Providing the individual impact of AI in % in the main text could be of interest with regards to the scope of the manuscript.

Thank you very much for your suggestion. We added individual impact of AI in % in the main text.

Suggested improvements

I suggest to include the option to replace packaged air conditioners by heat pump units, especially for HEE and NZE buildings. Another important element could be the extension of the present study to commercial and institutional buildings at large, not only medium sized office buildings; nonetheless, this aspect is optional since it might require significant changes. However, the authors could provide more information about the methodology to come up with such an analysis.

Authors' response:

Thank you for your review. We greatly appreciate your suggestion to extend the study beyond medium-sized office buildings to include commercial and institutional buildings. While our focus was on medium-sized office buildings in this study, we acknowledge the importance of considering a broader range of building types and we conducted the study with this extension in mind. We believe that our methodologies can be applied to different commercial building types with appropriate adjustments. This will involve adjusting various parameters and considering the specific characteristics and energy consumption patterns of each building category. We plan to explore these aspects in our future work and expand the analysis to encompass a wider range of commercial and institutional buildings. By doing so, we aim to develop a comprehensive understanding of the energy savings potential and carbon reduction opportunities across diverse building types.

Question 5:

Clarity and context

The manuscript is well written and easy to follow. A short description of the paper organization could be provided after line 62 to help the reader better follow the approach. Some typos were found throughout the text and should be corrected. The y axis label in Figure 2 shows “energy”, not “energy”; the y axis label in Figure 2 – right should be modified to “Energy savings from building design and construction”. In Table 2, there is an extra row in “operation”, which should be removed. At line 170, “t” should be corrected to “it”. “Subsides” at line 197 might be corrected by “subsidies”. In Supplementary Table 11, 2021 is given instead of 2020.

Authors' response:

Thank you for your valuable feedback. We appreciate your positive remarks about the overall clarity and organization of the manuscript. We have taken note of the suggested improvements and made the necessary revisions.

To enhance the reader's understanding, we have added a brief description of the paper's organization after line 62. This addition will provide a clearer structure and aid in following the approach presented in the manuscript.

Regarding the identified typos and errors, we have carefully reviewed and corrected them. The y-axis label in Figure 2 has been corrected to "energy" instead of "energy," and the y-axis label

in Figure 2 - right has been modified to "Energy savings from building design and construction" as suggested.

We have also addressed the extra row in the "operation" section of Table 2 and removed it accordingly. Additionally, the typo "tt" at line 170 has been corrected to "it," and "Subsides" at line 197 has been amended to "subsidies" for accuracy. We have also rectified the year in Supplementary Table 11 to reflect 2020 instead of 2021.

We appreciate your meticulous review, which has contributed to improving the quality of our manuscript.

Question 6:

References

Overall, the manuscript references previous literature appropriately. However, additional references could be provided to support the occupancy influence energy savings of 15-20% (line 147) and the information provided in Table 2 (AI application examples with energy and cost saving potential). Numbers given in lines 306-320 could also be broken down for each assumption, instead of being lumped together in the line 307. Additional work could be performed to break down long lists of references (e.g. 10-18 line 46; 19-33 line 50; 26,28,29,38,39 line 193; 24-26,28-29,34,39,43-47 line 307) whereas references could be added to support the method section (no references are provided).

Authors' response:

Thank you for your feedback on the references in our manuscript. We appreciate your suggestions to further support the information provided in the paper.

To strengthen the evidence for the occupancy influence energy savings of 15-20% (line 147) and the examples presented in Table 2, we have included additional references that provide relevant research and data in these areas. These additional references will help substantiate the claims and improve the overall credibility of the paper.

Regarding the breakdown of numbers in lines 306-320, we have revised the text to provide a more detailed breakdown of the assumptions and their corresponding sources.

We have also taken your advice to break down long lists of references throughout the manuscript. By separating the references and including specific citations, we aim to provide clearer sources of information.

Furthermore, we have added references to support the methodology section, ensuring that the appropriate sources are cited to provide a comprehensive understanding of the methods employed in the study.

We appreciate your valuable input, which has helped us strengthen the references in our manuscript and improve its overall quality.

Reviewer #3:

Remarks to the Author:

Overall, the I am satisfied with the changes done.

Reviewer #4:

Remarks to the Author:

This reviewer agrees the another reviewer's comments that the developed AI-based method for reducing building energy consumption emission cannot show great value in theory and practicability. It cannot meet the high requirement of this journal. The biggest problem is the novelty and the value of conclusions. It is hard to see the great value of AI in the targeted field. Particularly, the data are mostly the simulation data, and the validation and implementation is very weak. The questions in the last cycle of the review are not well addressed. This reviewer recommends the rejection of the paper.

Reviewer #6:

Remarks to the Author:

Overall, the authors have made a significant improvement of the paper by addressing the reviewers' comments. However, still the paper needs more improvement, as explained below:

1) The literature review of this article is very terse. Many recent studies have been missed and not discussed. The authors should elaborate more on this by discussing the following studies: Assessment of Building Energy Simulation Tools to Predict Heating and Cooling Energy Consumption at Early Design Stages; Artificial intelligence based anomaly detection of energy consumption in buildings: A review, current trends and new perspectives; AI-big data analytics for building automation and management systems: a survey, actual challenges and future perspectives; AutoBPS: A tool for urban building energy modeling to support energy efficiency improvement at city-scale; Neural network-based model predictive control system for optimizing building automation and management systems of sports facilities; Using urban building energy modeling data to assess energy communities' potential; Next-generation energy systems for sustainable smart cities: Roles of transfer learning; Data-informed building energy management (DiBEM) towards ultra-low energy buildings; Forecasting heating and cooling loads in residential buildings using machine learning: a comparative study of techniques and influential indicators; Performance and energy optimization of building automation and management systems: Towards smart sustainable carbon-neutral sports facilities; An innovative deep anomaly detection of building energy consumption using energy time-series images;

2) The limitations and drawbacks of the proposed method should be discussed in the Conclusion before deriving future work.

Response to Reviewers

We would like to express our sincere gratitude to the reviewers for your valuable comments and suggestions. Your feedback has greatly contributed to the improvement of the manuscript. We have carefully considered each comment and made necessary modifications to address the raised concerns. All the revisions have been clearly highlighted in the revised manuscript for easy reference. Please refer to our point-to-point responses below for a detailed overview of the changes implemented.

Reviewer #1 (Remarks to the Author):

The article is well prepared. It covers aspects within the area of Journal. In my opinion no further changes are needed.

Authors' response:

Thank you for your positive feedback and for considering the article well-prepared. We appreciate your assessment that the article covers relevant aspects within the scope of the journal. Your opinion that no further changes are needed is valuable and reassuring. We will address any remaining comments or suggestions from the other reviewers and make necessary revisions accordingly. Your feedback is greatly appreciated.

Reviewer #2 (Remarks to the Author):

This paper developed the AI-based method for reducing building energy consumption and emission. However, it cannot show great value in theory and practicability. It cannot meet the high requirement of this journal. The biggest problem is the novelty and the value of conclusions. It is hard to see the great value of AI in the targeted field. The comments are given below:

Question1:

1. In fact, AI has been widely adopted in green building design. There are many successful applications with the help of various intelligent algorithms. Hence, it is improper to say that AI in improving energy efficiency and reducing carbon emissions has not been studied systematically. In this regard, the research gap raised by the author is doubtful. To compare your work with other existing ones, what are the merits of your work? What improvement have you made?

Authors' response:

Thank you for your valuable feedback. We acknowledge that there have been successful applications of AI in green building design. However, our intention was to highlight the lack of studies that comprehensively assess the overall impact of AI on energy efficiency and carbon emissions reduction. While individual case studies exist, their focus is often limited to specific aspects of building performance such as design, construction, or operation. Consequently, these studies often yield a wide range of potential energy savings, ranging from 2% to 60%. This wide variation can be less informative and challenging for decision-makers seeking clearer and more actionable insights.

The novelty of our work lies in two key aspects. Firstly, we provide a holistic perspective by considering all four key categories: equipment, occupancy influence, control and operation, and design and construction. This comprehensive approach sets our research apart from previous studies. Secondly, we analyze the impact of AI on a national scale using technology adoption and building stock modeling. Our findings indicate that AI can significantly contribute to energy consumption and carbon emissions reduction by reducing the cost of technology adoption at scale.

We realized that we might have not clearly articulate these novelties in our original submission. We appreciate your feedback, which has prompted us to emphasize the unique contributions of our research.

Question2:

2. Can you provide the detail of the efficient policy implementation? This is the key content of your work. However, it is unclear. Other readers could be hard to follow and repeat your work. What is the policy? How can we obtain the policy and carry out it? Does it a data-driven policy making process? How can you test the effectiveness of the policy?

Authors' response:

Thank you for raising important questions regarding the details of our efficient policy implementation. We appreciate the opportunity to provide further clarification.

The major policy measures employed in our scenario include the investment on R&D, deployment of cost-effective technologies including additive manufacturing, promotion of efficiency technologies, implementation of building codes and energy efficiency standards, incentives, subsidies, financial assistance, and government-funded programs. These policies are consistent with the pathway to the Long-Term Strategy of the United States, which aims for 100% clean electricity by 2035 and net-zero greenhouse gas emissions by 2050.

The policy scenario in our study encompasses four main pathways:

1. Decrease in the cost premium for highly energy-efficient (HEE) or net-zero energy (NZE) medium office buildings through investment on R&D and deployment of cost-effective technologies including additive manufacturing, incentives, subsidies, financial assistance, and government-funded programs.
2. Increase in the retrofit share of existing commercial buildings to improve their energy efficiency through building codes and energy efficiency standards, incentives, subsidies, financial assistance, and government-funded programs.
3. Increase in the share of new commercial buildings that achieve net-zero energy status until the maximum allowed share is reached for each region or climate zone through investment on R&D and deployment of cost-effective technologies including additive manufacturing, building codes and energy efficiency standards, incentives, subsidies, financial assistance, and government-funded programs.
4. Increase in the share of retrofitted commercial buildings that achieve net-zero energy status until the maximum allowed share is reached for each region or climate zone through building codes and energy efficiency standards, incentives, subsidies, financial assistance, and government-funded programs.

Specific details regarding these pathways can be found in Supplementary Table 9-12 of our manuscript.

While our policy scenario is based on relatively conservative assumptions, taking into account the timeline and economically justified maximum allowed shares for new and existing buildings, we also considered variations among different regions and the differences between residential and commercial applications. This approach allows us to generalize our results to the entire commercial buildings sector.

Although our policy-making process involves setting assumptions and pathways based on factors such as construction costs and renewable energy potential data, we acknowledge that it is not strictly a data-driven policy-making process. Instead, our policies are aligned with national and subnational policies already adopted such as in California and New York.

We appreciate your feedback and have addressed these points in our revised manuscript to provide a clearer understanding of the efficient policy implementation in our study. Please refer to paragraphs 3-5 in Section 3.

Question3:

3. From the method part, the reviewer does not agree that this is an AI method. Instead, it can only be regarded as a mathematical modeling method. How can we see the intelligence from

your method? Are there any parts with the function of prediction and optimization in your method? This paper is not innovative enough.

Authors' response:

Thank you for your valuable feedback.

We would like to clarify that the main objective of our study is not to focus on a specific AI method, but rather to generally explore the potential contribution of AI in enhancing energy efficiency and reducing carbon emissions. Our approach primarily utilizes mathematical modeling and engineering analysis to investigate how AI can help reduce construction time and costs, leading to the increased adoption of high-performance buildings. Our focus is on the broader potential of AI in driving improvements in energy efficiency and carbon reduction through economic and market effects.

Having said this, our methodology does leverage AI-related techniques indirectly. By considering various factors such as occupancy influence, control and operation, and design and construction, we incorporate predictive and optimization aspects in our analysis. These elements contribute to the overall goal of reducing energy consumption and carbon emissions by identifying cost-effective technology adoption strategies.

We appreciate your feedback regarding the level of innovation in our paper. While our study may not introduce entirely new AI methods, it provides valuable insights into the potential benefits of AI in achieving energy efficiency and carbon emissions reduction at scale. We will make sure to clarify these points and highlight the innovative aspects in our revised manuscript.

Question4:

4. Is the method generalized enough to extend to other cases? This journal hopes the authors to provide some high-level conclusions, which cannot limit to a certain case. In the current version, it lacks some useful and convincing results.

Authors' response:

Thank you for your feedback. We recognize the importance of a generalized method that can be applied to various cases. In our study, we utilized well-established engineering analysis techniques such as building energy simulation, technology adoption, and building stock modeling. These methodologies are widely accepted and commonly used in the US DOE's standard and regulation development processes.

Although we focused on a medium office building as an example in this paper, our methodology is not limited to this specific case. It can be extended and applied to other building types as well. By utilizing the same analytical framework and considering the unique characteristics of different building types, our method can provide valuable insights and high-level conclusions applicable to a broader range of cases.

Thank you for pointing out that we did not explicitly state the generalizability of our method in the current version. Your feedback has prompted us to clarify this aspect in our revised manuscript, ensuring that our conclusions are useful and convincing beyond the specific case presented.

Question5:

5. Does the data used in this paper come from the EnergyPlus simulation tool? Is it reliable? The implementation of the tool will influence the quality of data. Do you have some real data to be used?

Authors' response:

Thank you for your question regarding the reliability of the data used in our paper. The energy savings potential analysis is based on EnergyPlus simulations, which is a widely accepted and validated tool developed by the U.S. Department of Energy. EnergyPlus has been extensively used by researchers, practitioners, and regulatory agencies for building energy performance analysis.

In our study, we utilized prototype medium office buildings as examples, which are derived from DOE's Prototype Building Models and developed by the Pacific Northwest National Laboratory (PNNL). These prototype buildings (defined by ASHRAE standard 90.1) represent various building types and are well-calibrated based on real data. They have been used by the U.S. Department of Energy's Building Energy Codes Program for evaluating building energy codes and proposing code changes.

While we primarily relied on simulated data from EnergyPlus for our analysis, it is important to note that these simulations are based on validated models and real-world building characteristics. This provides a robust foundation for our findings.

Reviewer #3 (Remarks to the Author):

Overall, this is a well written manuscript with an interesting approach to estimate potential for AI. That said, this paper does require a bit of work.

Authors' response:

We appreciate your positive feedback on the manuscript and your recognition of our interesting approach to estimating the potential for AI. We have carefully reviewed your comments and suggestions, and we have made revision to address them. Please find our point-to-point response below, which outlines the specific changes we have made to improve the paper:

Question1:

- Title could use a modification, as this discusses more about the potential of AI.

Authors' response:

We appreciate your feedback on the title of our manuscript. Based on your suggestion, we have revised the title to better reflect the content and focus of the paper. The updated title is "Exploring the Potential of Artificial Intelligence in Reducing Energy Consumption and Carbon Emissions of Commercial Buildings at Scale". We believe that the new title accurately represents the discussion on the potential of AI. Thank you for bringing this to our attention, and we hope the revised title aligns more effectively with the scope of the paper.

Question2:

- The authors need to elaborate on each of their cases. For example, the authors explain that Case 1 increases the cooling efficiency by 20%. Case 2 increases the heating efficiency by 12%. Please justify how and why were such values selected? Moreover, how would the selection of percentages selected by the authors effect the overall analysis?

Authors' response:

Thank you for your valuable feedback and pointing out we did not provide sufficient details regarding the selection of percentages in each case. The values were chosen based on the difference of the energy performance of actual products and commercially available technologies.

To note, our analysis does not include the potential benefits of non-commercialized best available technologies, as the time needed for validation and commercialization is uncertain. We acknowledge that considering the non-commercialized technologies would likely result in higher energy saving potentials than our current estimates. Sensitivity analysis has been added in the later section to reflect the uncertainty from the energy savings potential.

We appreciate your input and have made the necessary revisions to address this concern. Thank you for bringing this to our attention.

Question3:

- Tables which summarize each energy evaluation would be beneficial to the readers.

Authors' response:

Thank you for your suggestion regarding the inclusion of tables summarizing each energy evaluation. We agree that tables would provide a clearer overview of the results and enhance the reader's understanding. In response to your comment, we have added relevant tables (Table 5-7) in the Supplementary Information document, which provide a comprehensive summary of the energy evaluations performed in our study. These tables include the key findings and metrics for each case analyzed. We believe that the addition of these tables will greatly benefit the readers in comprehending the results. Thank you for bringing this to our attention, and we appreciate your valuable input.

Question4:

- "We believe that AI can improve energy efficiency and reduce carbon emissions through two main approaches: (1) AI helps scale up the best available technologies and practices. Because, it can significantly help to scale up the technologies and speed adoption by reducing the construction and labor costs." Any examples of this happening? Please provide a reference(s) if possible.

Authors' response:

Artificial intelligence (AI) has gained significant attention and is being increasingly applied in various industries, including the construction industry. Its potential to reduce construction and

labor costs, mitigate risks for construction workers, and enhance their health benefits has been recognized. Construction companies are increasingly realizing the advantages of utilizing AI in different aspects of construction, ranging from design to the implementation of cutting-edge technologies and practices.

Due to the relatively recent emergence of AI applications in the construction industry, the existing literature in this field is still developing. A technical review article titled "Artificial intelligence in the construction industry: A review of present status, opportunities, and future challenges" (Abioye et al., 2021. J Build Eng 44, 103299) provides valuable insights into the subject. References 21-23 in the manuscript also provide some good information on this topic. In addition to technical resources, there are non-technical articles that discuss the transformative impact of AI in the construction industry. "How AI is transforming the construction industry" (<https://theconstructor.org/artificial-intelligence/how-ai-is-transforming-the-construction-industry/568678/>) and "Who says you can't teach an old dog new tricks? The case for AI in construction" (<https://www.forbes.com/sites/angelicakrystledonati/2023/03/07/who-says-you-cant-teach-an-old-dog-new-tricks-the-case-for-ai-in-construction/?sh=79bf45fd3654>) are two interesting reads that highlight the potential benefits and opportunities presented by AI. While the literature on AI applications in the construction industry is still evolving, these resources provide valuable insights into the promising potential of AI in reducing costs, improving construction processes, and enhancing worker well-being.

Question5:

- How well would such an analysis apply for other advance control systems? AI is shown to potentially save certain percentages. However, model predictive control (using RC models for example) may also potentially have a similar savings. Would your analysis differ for different types of advance controls? A sentence or two describing how the methodology within could be applied to other advanced control techniques/technologies would be beneficial

Authors' response:

Thank you for your insightful question. Our analysis primarily focuses on currently widely adopted control measures for energy savings in buildings. We acknowledge that other advanced control systems, such as Reinforcement Learning or Deep Learning models, may have the potential for additional savings beyond what is captured in our analysis.

The methodology presented in our study can serve as a foundation for evaluating the energy savings potential of different advanced control techniques and technologies. By incorporating the specific characteristics and benefits of these advanced controls into the analysis framework, it is possible to assess their impact on energy efficiency and carbon emissions reduction.

In future work, we plan to explore the application of these advanced control methods and technologies, such as RC models or deep learning, within the context of our analysis. This will allow us to provide a more comprehensive understanding of their potential contributions to energy savings and further improve the accuracy and applicability of our methodology.

We appreciate your suggestion and added a “Discussions” section on the application of the methodology to other advanced control techniques in the revised manuscript. Now the Discussions section reads as below:

Instead of concentrating on a specific AI technology, this paper utilizes engineering and energy simulation methods to quantify the potential impact of AI on enhancing energy efficiency and reducing carbon emissions in general. Enabled by advanced AI algorithms and techniques, data-driven modeling and decision-making approach can provide customized solutions and greatly enhance the adoption and implementation of high energy technologies at scale with low cost. To further improve the accuracy and applicability of the proposed methodology, the applications of advanced control models (such as deep learning or reinforcement learning) can further be explored by following the same framework in future work.

While this paper takes medium office as an example, the proposed methodology can be applied to different commercial building types with appropriate adjustments of the input parameters based on specific characteristics and energy consumption patterns of each building category. To develop a comprehensive understanding of AI’s energy savings potential and carbon reduction opportunities across diverse building types, future work can expand the analysis to encompass a wider range of commercial and institutional buildings.

Question6:

There are many formatting issues as well

- Main body text sometimes appears to be in different size fonts
- Tables heading are on one page while the table is on a second (e.g. Table 1)
- Tables are shown overlapped onto two pages
- Are lines 160 to 164 a note? normal paragraph?
- Figure 2 can be difficult to read. Please improve the quality
- Line 134, Case 17?

Spelling and grammar need to be checked

Authors' response:

Thank you for bringing the formatting issues to our attention. We apologize for the errors and appreciate your feedback. In response to your comments, we have carefully reviewed and revised the manuscript to address the formatting issues you mentioned. We have ensured consistent font sizes throughout the main body text, adjusted the placement of table headings to align with their respective tables, and resolved any overlapping issues with the tables. We have also clarified the formatting of lines 160 to 164. They are not normal paragraph, but notes in the Figure following Nature Communications' format. Additionally, we have improved the quality of Figure 2 to enhance its readability. Furthermore, we have corrected the reference to "Case 17" in line 134. It should be "Case 9". Lastly, we have conducted a thorough spell and grammar check to ensure the manuscript is free of any errors. We appreciate your attention to detail and thank you for helping us improve the quality of our manuscript.

Reviewer #4 (Remarks to the Author):

This study claims that AI can help reduce energy consumption and carbon emissions of buildings at scale and uses case studies for demonstration. This reviewer has the following comments for improvement.

Question1:

1. The role of AI is vague. This reviewer did not see clearly the contributions of AI in this study. It seems that the simulation techniques are used to solve the problem in this study. However, AI is not the same to the simulation tool. What are the data-driven modeling and decisions?

Authors' response:

Thank you for bringing up this point. We realized that we might have not clearly explain the role of AI in our study. We thus would like to clarify that the objective of this study was not exploring specific AI technologies in depth. Instead, our main objective is to investigate the potential of AI in increasing the market shares of high energy efficiency and net zero energy buildings. We believe that AI has the capability to accelerate construction processes, reduce costs, and enhance safety and health benefits for construction workers. Our study aims to shed light on how AI can facilitate these improvements in the construction industry.

The use of simulation techniques, such as energy modeling and analysis, is indeed an important aspect of our study. However, we also recognize the potential for AI to play a significant role in data-driven modeling and decision-making processes related to energy efficiency and carbon emissions reduction in buildings.

In our analysis, we primarily utilize engineering and energy simulation methods to estimate the energy savings potential. However, we believe that data-driven modeling and decision-making, enabled by AI techniques, can greatly enhance the adoption and implementation of energy-efficient technologies at scale. By leveraging AI algorithms, such as machine learning and optimization, it is possible to optimize building operations, predict energy consumption patterns, and make data-driven decisions for energy efficiency measures.

We appreciate your comment and will make sure to provide a clearer explanation of the potential role of AI in data-driven modeling and decision-making in the revised manuscript.

Question2:

2. The uncertainty in future predictions (e.g., 2050) of energy consumption and emission reduction is not considered in this research, which is not realistic in practices. This indicates that the results and findings are not reliable.

Authors' response:

Thank you for your valuable feedback. While our main focus is on evaluating the potential contribution of AI in scaling up best practice technologies and practices, we acknowledge the importance of addressing uncertainties in our research. In response to your suggestion, we have conducted a sensitivity analysis to provide a better understanding of the uncertainties involved in our future predictions. This analysis helps to enhance the robustness and reliability of our results and findings.

Question3:

3. In Fig. 4 and Fig. 5, what does “Frozen” mean? More explanations are needed.

Authors’ response:

Thank you for your valuable feedback. In our study, the term "Frozen" refers to a scenario where the market shares of three types of buildings (baseline buildings, high energy efficiency buildings, and net zero energy buildings) remain constant at the 2020 level throughout the future until 2050. This scenario serves as a counterfactual baseline for comparison in our analysis. We will make sure to provide a more detailed explanation in the figure captions to avoid any confusion.

Question4:

4. The scenarios with and without AI are not clearly presented.

Authors’ response:

Thank you for your valuable feedback. We added a new Supplementary Table 5 to explain the definitions of each scenarios. All the pathways related to the scenarios are listed in the supplementary Tables 11-14. In our study, the key distinction between the scenarios lies in the role of AI in accelerating the design, implementation, and operation of energy efficiency technologies throughout the entire building lifecycle. Additionally, the scenarios with AI incorporate an additional 10% reduction in construction and labor costs compared to the scenarios without AI. This cost reduction enhances the competitiveness and subsequent adoption of energy efficiency technologies throughout the building lifecycle.

To elaborate on the implementation of AI in our study, we utilized discrete choice modeling to capture the impact of AI on cost reduction, which, in turn, drives the increased adoption of high energy performance (low EUI) buildings and net zero energy buildings. With the inclusion of AI, the adoption rate of high energy efficiency buildings and net zero energy buildings is higher due to the reduction in construction time and costs.

In summary, the scenario with AI leads to a higher market share of high energy efficiency buildings and net zero energy buildings over time compared to the scenario without AI. This trend continues until the market share of net zero energy buildings reaches its maximum share.

Reviewer #5 (Remarks to the Author):

Thank you for providing such detailed comments on our paper. Your feedback has been immensely valuable and has greatly contributed to the improvement of the paper's quality. We truly appreciate the effort and thoughtfulness you put into reviewing our work.

Your insightful comments have inspired us to make significant revisions to address the specific points you raised. We carefully considered your suggestions and incorporated them into the revised manuscript. Your input has undoubtedly strengthened the overall content and clarity of our paper.

Once again, we extend our sincere gratitude for your thorough review and valuable insights. Your contribution has been instrumental in enhancing the quality of our work. Please find our point-to-point response below for your review.

Key results

The manuscript deals with the role of artificial intelligence to improve energy efficiency and reduce carbon dioxide emissions in U.S. medium office buildings and aims to quantify potential reductions for different 2020-2050 scenarios. Alongside with energy policy and low carbon power generation, AI could help reduce up to energy use by 47% and GHG emissions by 93%.

Validity

The validity and robustness of the data interpretation and conclusions are in general appropriate. However, there are several flaws, which should be addressed.

Question 1:

One of the main drawbacks is that the authors focus only on medium-sized office buildings to evaluate energy consumption and carbon dioxide reduction in U.S. buildings as stated in lines 19-20. Reasons for considering medium-sized office buildings should be provided, along with suitable statistics to support this approach such as the market share of medium-sized office buildings as well as their contribution to the total building energy use and/or GHG emissions in buildings.

Authors' response:

Thank you for your comment. Although our paper aims to establish a methodology that can be widely applied to assess AI impact to building energy consumption reductions, we acknowledge the importance of providing a clear rationale for focusing on medium-sized office buildings in our study. We agree that it would be beneficial to include suitable statistics to support this approach and highlight the significance of medium office buildings in the context of total building energy use and greenhouse gas emissions.

In our revised manuscript, we will include the statistics to show the importance of medium office. It dominates the energy consumption of all office buildings.

Thank you for bringing this to our attention, and we appreciate your feedback in helping us strengthen our study.

Question 2:

Another important flaw is how AI can be applied to improve building performance. Several applications are provided, including equipment, imperfect design, imperfect construction, subpar controls and operation. In some of the proposed examples, AI could be helpful but it might not be required. For instance, a rule-based FDD with no AI could possibly provide similar savings compared to an AI-based FDD whereas MPC could rely on white-box or grey-box models, rather than AI models, while achieving similar energy savings. In this specific case, the AI term encompasses improved controls and operation in general, not just AI-based controls and operation, which could be a bit misleading about the potential of AI. The wording could be modified accordingly. Similar remarks could be done for the other applications.

Authors' response:

Thank you for your insightful comment. We appreciate your perspective on the potential application of AI in improving building performance and the distinction between AI-based approaches and other techniques such as rule-based FDD and model predictive control.

In our revised manuscript, we will clarify the definition of AI and emphasize that our focus is on the potential of AI to enhance the adoption of energy-saving measures at a large scale, rather than analyzing one specific AI approach. We agree that there are various techniques, including rule-based methods and model-based control, that can contribute to energy savings without explicitly relying on AI models. We believe that AI could help speed up the adoption of energy efficiency technologies and advance the development of high-performance buildings by reducing construction time and costs, while also enhancing the health and well-being of construction workers.

By modifying the wording accordingly, we aim to provide a more accurate representation of the role of AI and its potential in improving building performance, while acknowledging the effectiveness of other approaches. We appreciate your valuable input in helping us refine our paper.

Question 3:

Finally, the study focusses on buildings with packaged air conditioning units equipped with natural gas furnace and the authors evaluated potential increases in cooling and heating efficiency in Supplementary Table 2. However, alternatives to such a conventional system have not been investigated while some options could significantly affect study results. In the context of 2050 emission targets, heat pumps might offer more potential for energy and cost savings, might be more suitable especially for HEE and NZE buildings while possibly allowing carbon neutral buildings by 2050.

Authors' response:

Thank you for your valuable comment. We agree that alternative systems, such as heat pumps, have the potential to significantly impact energy and cost savings, particularly in the context of

meeting 2050 emission targets and achieving carbon-neutral buildings. We appreciate your suggestion to include heat pump options in our analysis.

In the revised manuscript, we have included one new scenario (Case 9) under the Equipment efficiency category specifically addressing heat pump solutions. It considers the replacement of packaged air conditioning units with heat pumps for space heating. We have modeled and estimated the technical energy savings associated with this scenario.

By incorporating these heat pump scenarios, we aim to provide a more comprehensive analysis of potential energy savings and further explore the role of heat pumps in achieving energy efficiency and carbon reduction goals. We thank you for bringing this important aspect to our attention and contributing to the enhancement of our study.

To ensure the generalizability of our conclusions beyond medium office buildings, we considered the challenges faced by large commercial buildings, particularly in cold or very cold climate zones that require significant heating capacity within short time frames. Given the limited availability and higher costs of current heat pump systems for such requirements, we included natural gas heating as a baseline in our study. However, we recognize that advancements in heat pump technologies may make them more competitive in the future. In our future research, we plan to investigate the maximum technical and economic potential of heat pump systems.

For this study, we took a conservative approach by adopting realistic assumptions and pathways that consider the limitations of existing heat pump products and the substantial heating needs of certain large commercial buildings. Our primary objective is to assess the contribution of AI in achieving energy savings and reducing carbon emissions in commercial buildings by promoting the adoption of high energy efficiency and net zero buildings. We believe that our methodology, with its conservative assumptions, can be applied to various types of commercial buildings with appropriate adjustments, allowing for the potential impact of AI to be evaluated across the sector.

Significance

The manuscript provides reasonable estimates of the potential of AI for supporting the decarbonization of buildings to meet 2050 emission target. Such a piece of work helps promote the need for advanced tools targeting improved design, construction, operation and maintenance by assessing associated savings, and could support the development of new policy promoting the use of AI tools and data-driven solutions in general.

Question 4:

Data and methodology, analytical approach

The proposed methodology to evaluate how AI could help reduce energy consumption and carbon emissions in buildings is interesting and seems correct. Nonetheless, some clarifications are required to better understand the approach. Firstly, the method to estimate potential savings related to AI is confusing. In the paragraph at lines 66-72, the energy use intensity is provided for median U.S. office (167 kWh/m²) and for low energy verified buildings (57 kWh/m²). However, it is not clear how this value of 57 kWh/m² is used afterwards in the manuscript. Indeed, the proposed approach intends to estimate savings from a baseline (the median U.S. office) and might not need to refer to the low energy verified building EUI. In addition, Figure 3 also shows the median of zero (not low) energy verified building in the U.S.

Authors' response:

Thank you for your valuable feedback. The primary objective of this study is not to investigate specific AI technologies. Instead, we aim to explore how AI can contribute to reducing energy consumption and carbon emissions by decreasing overall construction time and costs. More specifically, we investigate two aspects: (1) the acceleration of design, implementation, and operation and maintenance (OM) of energy efficiency technologies throughout the building lifecycle, and (2) the reduction of construction and labor costs. These outcomes would enhance the competitiveness and adoption of energy efficiency technologies in the overall building lifecycle. To achieve this, we employed discrete choice modeling in our study. This approach enables us to incorporate the cost reduction resulting from AI applications, which in turn amplifies the competitiveness and adoption of high energy efficiency (low EUI) buildings and net zero energy buildings.

Regarding the baseline EUI of low EUI buildings, it is set at 57 kWh/m². The EUI serves as the baseline for calculating energy savings when comparing low EUI buildings to high EUI buildings. For example, the unit energy savings achieved by adopting a low EUI building is $167 - 57 = 110$ kWh/m².

The “policy w/AI & LEPCG” allows to reach up to 95% emission savings. More information could be provided to describe how this low emission power generation can be achieved and why buildings cannot be carbon neutral (my guess is the natural gas furnace in packaged air conditioners).

Authors' response:

We appreciate your comments regarding the low emission power generation and the use of natural gas in certain areas. In our study, we assume a transition towards 100% clean electricity generation as part of our low emission power generation scenario.

Meanwhile, we have taken into consideration the unique heating requirements of large-scale commercial buildings in cold and very cold areas. These types of buildings often have high heating capacity demands within short timeframes, and currently, the widespread use of heat

pumps for such applications is still challenging. The heating energy usage patterns of these large-scale commercial buildings differ significantly from residential buildings and small-scale commercial buildings.

Considering these factors, we have accounted for a small share of natural gas use in our analysis. This is to reflect the current competitiveness of natural gas in large-scale commercial buildings, especially in the near to medium terms. We believe that natural gas will continue to play a role in meeting the heating demands of these specific building types.

We appreciate your input, and we will ensure that these points are clarified and further elaborated upon in the revised version of the manuscript.

The manuscript lacks information regarding the extrapolation from individual buildings to the whole country. The method should be presented along with the main assumptions (share of buildings per ASHRAE climate zone, etc.) and associated references.

Authors' response:

Our study uses DOE's prototype building models in each climate zone to capture typical energy usage patterns. The prototype building models are developed by the Pacific Northwest National Laboratory (PNNL). These prototype buildings (defined by ASHRAE standard 90.1) represent various building types and are well-calibrated based on real data. They have been used by the U.S. Department of Energy's Building Energy Codes Program for evaluating building energy codes and proposing code changes. While we focused on medium office buildings, our model incorporates representative and cost-effective technologies applicable to various commercial building types. As a result, we believe our findings can be generalized to most building types, given our relatively conservative assumptions and pathways.

The impact of AI algorithms is modelled with a factor to decrease in energy usage and cost in Eq. 7. This decrease might be put in perspective with side effects such as the CPU requirements and capacities resulting in increased electric power for the deployment of AI at large scale. I do not necessarily expect precise numbers but rather a discussion on the implication of using AI at large scale.

Authors' response:

Thank you for your valuable suggestion. We would like to note that our study focuses on the energy impact of the AI application, not generic impact of AI technologies. We do acknowledge that the specific hardware and infrastructure requirements for the large-scale deployment of AI should be an important topic for future studies.

Lastly, results given in Figures 4 and 5 show the impact of AI along with other measures such as new energy policy and low carbon power generation. Providing the individual impact of AI in % in the main text could be of interest with regards to the scope of the manuscript.

Thank you very much for your suggestion. We added individual impact of AI in % in the main text.

Suggested improvements

I suggest to include the option to replace packaged air conditioners by heat pump units, especially for HEE and NZE buildings. Another important element could be the extension of the present study to commercial and institutional buildings at large, not only medium sized office buildings; nonetheless, this aspect is optional since it might require significant changes. However, the authors could provide more information about the methodology to come up with such an analysis.

Authors' response:

Thank you for your review. We greatly appreciate your suggestion to extend the study beyond medium-sized office buildings to include commercial and institutional buildings. While our focus was on medium-sized office buildings in this study, we acknowledge the importance of considering a broader range of building types and we conducted the study with this extension in mind. We believe that our methodologies can be applied to different commercial building types with appropriate adjustments. This will involve adjusting various parameters and considering the specific characteristics and energy consumption patterns of each building category. We plan to explore these aspects in our future work and expand the analysis to encompass a wider range of commercial and institutional buildings. By doing so, we aim to develop a comprehensive understanding of the energy savings potential and carbon reduction opportunities across diverse building types.

Question 5:

Clarity and context

The manuscript is well written and easy to follow. A short description of the paper organization could be provided after line 62 to help the reader better follow the approach. Some typos were found throughout the text and should be corrected. The y axis label in Figure 2 shows “energy”, not “energy”; the y axis label in Figure 2 – right should be modified to “Energy savings from building design and construction”. In Table 2, there is an extra row in “operation”, which should be removed. At line 170, “tt” should be corrected to “it”. “Subsides” at line 197 might be corrected by “subsidies”. In Supplementary Table 11, 2021 is given instead of 2020.

Authors' response:

Thank you for your valuable feedback. We appreciate your positive remarks about the overall clarity and organization of the manuscript. We have taken note of the suggested improvements and made the necessary revisions.

To enhance the reader's understanding, we have added a brief description of the paper's organization after line 62. This addition will provide a clearer structure and aid in following the approach presented in the manuscript.

Regarding the identified typos and errors, we have carefully reviewed and corrected them. The y-axis label in Figure 2 has been corrected to "energy" instead of "enery," and the y-axis label

in Figure 2 - right has been modified to "Energy savings from building design and construction" as suggested.

We have also addressed the extra row in the "operation" section of Table 2 and removed it accordingly. Additionally, the typo "tt" at line 170 has been corrected to "it," and "Subsides" at line 197 has been amended to "subsidies" for accuracy. We have also rectified the year in Supplementary Table 11 to reflect 2020 instead of 2021.

We appreciate your meticulous review, which has contributed to improving the quality of our manuscript.

Question 6:

References

Overall, the manuscript references previous literature appropriately. However, additional references could be provided to support the occupancy influence energy savings of 15-20% (line 147) and the information provided in Table 2 (AI application examples with energy and cost saving potential). Numbers given in lines 306-320 could also be broken down for each assumption, instead of being lumped together in the line 307. Additional work could be performed to break down long lists of references (e.g. 10-18 line 46; 19-33 line 50; 26,28,29,38,39 line 193; 24-26,28-29,34,39,43-47 line 307) whereas references could be added to support the method section (no references are provided).

Authors' response:

Thank you for your feedback on the references in our manuscript. We appreciate your suggestions to further support the information provided in the paper.

To strengthen the evidence for the occupancy influence energy savings of 15-20% (line 147) and the examples presented in Table 2, we have included additional references that provide relevant research and data in these areas. These additional references will help substantiate the claims and improve the overall credibility of the paper.

Regarding the breakdown of numbers in lines 306-320, we have revised the text to provide a more detailed breakdown of the assumptions and their corresponding sources.

We have also taken your advice to break down long lists of references throughout the manuscript. By separating the references and including specific citations, we aim to provide clearer sources of information.

Furthermore, we have added references to support the methodology section, ensuring that the appropriate sources are cited to provide a comprehensive understanding of the methods employed in the study.

We appreciate your valuable input, which has helped us strengthen the references in our manuscript and improve its overall quality.

Reviewers' Comments:

Reviewer #4:

Remarks to the Author:

The main problem of this article is that the role of AI is not rightly used. The application of AI is not clear and confusing. The authors did not reply directly to the questions raised. They used many qualitative sentences to describe the potential of AI use, instead of the quantitative results. As the results, the conclusions are not confusing. I think this can not achieve the high standard of Nature Communications.

Reviewer #6:

Remarks to the Author:

The authors have addressed all the reviewers' comments; no further revision is needed.

Response to Reviewers

Reviewer #4 (Remarks to the Author):

The main problem of this article is that the role of AI is not rightly used. The application of AI is not clear and confusing. The authors did not reply directly to the questions raised. They used many qualitative sentences to describe the potential of AI use, instead of the quantitative results. As the results, the conclusions are not confusing. I think this can not achieve the high standard of Nature Communications.

Authors' response:

We acknowledge that the reviewer might have a different view or definition of AI compared to our understanding. We would like to clarify that the objective of this study was not exploring specific AI technologies in depth. Instead, our main objective is to investigate the potential of AI in increasing the market shares of high energy efficiency and net zero energy buildings. We believe that AI has the capability to accelerate construction processes, reduce costs, and enhance safety and health benefits for construction workers. Our main focus is on evaluating the potential contribution of AI in scaling up best practice technologies and practices.

We conducted quantitative analysis and sensitivity analysis to study the AI contributions. Please refer to the Results section. The scenario with AI leads to a higher market share of high energy efficiency buildings and net zero energy buildings over time compared to the scenario without AI. This trend continues until the market share of net zero energy buildings reaches its maximum share. Based on your comments, we have added several new analyses for Figure 4 and Figure 5, highlighting the energy savings and carbon emission reductions with AI vs without AI. AI adoption could reduce energy consumption and carbon emissions by approximately 8% to 19% in 2050. Combining AI with energy policy and low-carbon power generation could approximately reduce energy consumption by 40% and carbon emissions by 90% in U.S. buildings compared to business-as-usual scenarios in 2050.

Reviewer #6 (Remarks to the Author):

The authors have addressed all the reviewers' comments; no further revision is needed.

Authors' response:

Thank you for your positive feedback and for considering the article well-prepared. Your opinion that no further changes are needed is valuable and reassuring. We will address any remaining comments or suggestions from the other reviewer and make necessary revisions accordingly. Your feedback is greatly appreciated.